# Lake Ice Break-Up in Greenland: Timing and Spatio-Temporal Variability

Christoph Posch[1], Jakob Abermann[1], Tiago Silva[1]

[1]Institute of Geography and Regional Science, University of Graz, Graz, A-8010, Austria

*Correspondence to*: Christoph Posch (christoph.posch@edu.uni-graz.at)

**Abstract.** Synthetic aperture radar (SAR) data from the Sentinel-1 (S1) mission with its high temporal and spatial resolution allows for an automated detection of lake ice break-up timings from surface backscatter differences across South (S), Southwest (SW) and Northwest (NW) Greenland (< 71° N latitude) during the period 2017 to 2021. Median break-up dates of the 563 studied lakes range between 8 June and 10 July, being earliest in 2019 and latest in 2018. There is a strong correlation

between break-up date and elevation, while a weak relationship with latitude and lake area could be observed. Lake-specific median break-up timings for 2017-2021 increase (i.e., are later) by 3 days per 100 m elevation gain. When assuming an earlier break-up timing of 8 days which corresponds to the observed median variability of ± 8 days, the introduced excess energy due to a changing surface albedo from snow covered ice surface to water translates to melting 0.4 ± 0.1 m thick ice at the melting point or heating up a water depth down to 35 ± 3 m by 1 K across the entire surface area of each respective lake. Upscaling

the results to 100486 lakes across the regions S, SW and NW which correspond to 64.5 % of all lakes or 62.1 % of the overall lake area in Greenland yields an estimate of 1.8 * $10^6$ TJ additional energy input. This translates to melting 5.8 Gt ice at the melting point or warming 432.3 Gt water by 1 K.

## 1 Introduction

Lake ice plays an important role in biological, chemical and physical processes of cold region freshwater (Duguay et al., 2015).

Freshwater ice in the Arctic and its response to climate change have a variety of effects on hydrologic, ecological, and socio-economic systems (Prowse et al., 2011) while climate change being one of the most severe threats to global lake ecosystems (Woolway et al., 2020). The duration of lake ice controls the seasonal heat budget of lakes and may have an effect on both regional climate and weather events (Brown and Duguay, 2010; Duguay et al., 2015). The timings of lake ice freeze-up and break-up, i.e., lake ice phenology, are relevant climate indicators and can be useful for monitoring environmental changes

(Adrian et al., 2009; WMO et al., 2023a). Therefore, lake ice is a parameter of the Essential Climate Variable (ECV) "lakes" and included in monitoring programs such as the World Meteorological Organization (WMO) Global Climate Observing System (GCOS) (WMO et al., 2023a) and the European Space Agency (ESA) Climate Change Initiative (CCI) (Climate Change Initiative Lakes, 2023). The scientific value of lake research and the important role of lakes for humans makes them incorporated in the United Nations' Sustainable Development Goals No. 6 (Clean Water and Sanitation) and No. 13 (Climate

Action) (United Nations, 2023) and an essential component of the United Nations Framework Convention on Climate Change (UNFCCC) and the Intergovernmental Panel on Climate Change (IPCC) (Woolway et al., 2020).

Lake ice freeze-up and break-up are results of energy surplus or deficit in the energy balance of the lake. The energy exchanges between the ice cover or water surface and the atmosphere are mainly determined by air temperature, precipitation, wind and radiation. The seasonal changes in solar radiation, however, are the main influence for the overall energy availability to form and decay lake ice cover (Brown and Duguay, 2010). Both linear and non-linear relation between lake ice break-up timing and air temperature have been established, while stronger correlations with latitude were identified compared to elevation (Magnuson, et al., 2000; Weyhenmeyer et al., 2004; Williams et al., 2004; Duguay et al, 2006; Korhonen, 2006; Williams and Stefan, 2006; Brown and Duguay, 2010; Jeffries et al., 2012; Imrit and Sharma, 2021).

Satellite remote sensing provides the necessary means to increase the spatial coverage and temporal frequency of ground-based observations of lake ice phenology which have been globally declining since the 1980s (Duguay et al., 2015). Synthetic aperture radar (SAR) backscatter exhibits differences between water and ice due to dielectric properties of the materials (Unterschultz et al., 2009) and therefore allows for identifying the phenological state of the lake ice cover. Several studies investigated the evolution, characteristics and phenology of freshwater ice such as river ice (e.g., Lindenschmidt et al., 2011; Stonevicius et al., 2022) and lake ice (e.g., Wang et al., 2018; Murfitt and Duguay, 2020; Tom et al., 2020; Siles et al., 2022) from radar imagery, while we are not aware of a comprehensive study on lake ice break-up timing across Greenland.

In this study, we explore the potential of utilizing Sentinel-1 (S1) SAR data for identifying temporal and spatial variations of lake ice break-up across S, SW and NW Greenland between 2017 and 2021 and assess its latitudinal and vertical gradients. Peripheral lakes in Greenland, i.e., lakes excluding supra- and proglacial lakes, make up for approximately 0.7 % of the overall land area or approximately 3 % of the unglaciated area. Therefore, we aim to quantify the additional energy input by estimating excess radiation and energy for a potential earlier lake ice break-up timing from the observed lake ice break-up variabilities.

## 1.1 Climate in Coastal Greenland

Greenland extends for approx. 23° of latitude, with temperature, precipitation and consequently mass balance rates varying considerably across latitudes and coasts (Westergaard-Nielsen et al., 2020; Hanna et al., 2021; Mankoff et al., 2021; Slater et al., 2021, Box et al., 2023). Due to the semi-permanent Icelandic Low and the rocky landscape, the Southeast coast receives particularly high amounts of precipitation (e.g., Ettema et al., 2010; Fettweis et al., 2017). As precipitation rates greatly decrease northward, North Greenland is classified as a polar desert with very shallow snow cover that quickly disappears in the warm season. Temperature also tends to decrease with latitude, related to snow and radiation conditions. However, other factors shape the coastal climate such as prominent ocean currents (e.g., East Greenland and North Atlantic current) as well as sea ice conditions (Westergaard-Nielsen et al., 2020). The West and East coasts also exhibit different topographic features, from a topographically complex Southeast contrasting with generally rather gentle slopes in Southwest or North Greenland (Karami et al., 2017). Nevertheless, both the East and West coasts comprise diverse fjord systems impacting regional climate and local wind conditions. Consequently, the leeward side of these inland mountain systems receive reduced precipitation.

Such coast-inland gradients are therefore complex, influencing the distribution of permafrost and freshwater systems (Westergaard-Nielsen et al., 2018; Abermann et al., 2021). While several studies on accumulation rates and rainfall exemplify

the general East-West gradient in precipitation focusing on the Greenland Ice sheet (GIS) (e.g., Shen et al., 2012; Koenig et al., 2016, Box et al., 2023), Bales et al. (2009) include and highlight coastal variability of snow accumulation. In-situ and remote sensing data as well as polar-adapted climate models point to a Greenland-wide warming in recent decades, particularly in summer (e.g., Westergaard-Nielsen et al., 2018; Jiang et al., 2020; Hanna et al., 2021). Part of this warming is attributed to more frequent and intense anti-cyclonic conditions in the vicinity of Greenland, leading to advection of relatively warm air

masses from low latitudes. Silva et al. (2022) showed that the warming applies to different circulation conditions. As a consequence of atmospheric warming, the ratio of liquid to total precipitation has increased in coastal areas particularly during summer (e.g., Huai et al. 2022; van der Schot et al. 2023). The Arctic Amplification is more pronounced during the cold season, with coastal temperature warming along the West coast linked with reduced sea ice in the Baffin Bay (e.g., Ballinger et al., 2021).

## 1.2 Background and Related Studies Using SAR for Studying Lake Ice Cover

The transmitted pulse of SAR systems interacts with the Earth surface and only a portion of it is backscattered to the receiving antenna. The amplitude and phase of the backscattered signal depends on the physical (i.e., geometry, roughness) and electrical properties (i.e., permittivity) of the imaged object (Moreira et al., 2013). The reflection, transmission and absorption of the radar beam at lake ice is governed by the (combination of) interactions with water, ice, snow and air. The differences in the

amplitude of the backscattered signal between ice and water due to the influence of the electrical properties (Unterschultz et al., 2009) can be utilized to identify the phenological state of the ice cover of lakes.

Using satellite data for studying lake ice possesses several advantages over ground observations in terms of data availability and accessibility. Ground observations of lake ice may be limited due to access to remote and unpopulated areas, safety hazards during freeze-up and break-up periods. Satellite observations are independent from these restrictions and offer a relatively

rapid, lower-cost and spatially broader way of obtaining data (Siles et al., 2022). Radar is independent from daylight and weather conditions and S1 SAR data is available and accessible at high spatial and temporal resolutions (Sentinel-1 SAR User Guide, 2023). Lake ice studies from remote sensing can produce results and extrapolate field measurements across large spatial scales as opposed to field studies based on a small number of ground observations. However, field observations are pivotal for validation purposes. (Murfitt and Duguay, 2021).

Wang et al. (2018) utilized dual polarized RADARSAT-2 imagery for a semi-automated, pixel-by-pixel ice/water classification at Lake Erie and provided an overall accuracy of up to 90.4 %. Using a deep learning network, Tom et al. (2020) conducted a pixel-based lake ice phenology classification from S1 SAR data for three alpine lakes in Switzerland and found accuracies well above 90 %. They demonstrated that the phenological state of the lake ice cover of non-transition days (i.e., ice/snow or water) can be identified confidently. Murfitt and Duguay (2020) utilized S1 high-density time series data to monitor ice

phenology of the High Arctic Lake Hazen (Canada) and demonstrated mean errors between 3 and 7 days for identifying the

timing of lake ice break-up. Since it has been demonstrated that lake ice phenology can be assessed using SAR data, we develop a dynamic numerical threshold to automatically identify the lake ice break-up timing from SAR backscatter across Greenland.

## 2 Data

### 2.1 Greenland Lake Inventory

The Greenland lake inventory (Styrelsen for Dataforsyning og Infrastruktur, 2023) includes 155870 peripheral lakes in Greenland ranging from $1.6 * 10^{-3}$ km$^2$ to 138 km$^2$. The vector data set is part of the data inventory Databoks Grønland (2023) and based on commercial satellite images with a resolution of 0.5 m primarily from summer months in the period from 2017 to 2021.

### 2.2 Sentinel-1 SAR

The Sentinel-1 mission consists of satellites S1A and S1B which acquire C-band SAR data with a center frequency of 5.407 GHz (Sentinel-1, 2023). Single polarized horizontal transmit/horizontal receive (HH) Level-1 ground range detected (GRD) data in both ascending and descending orbit acquired in Interferometric Wide (IW) swath mode with a swath width of 250 km is used in this study (Sentinel-1 SAR User Guide, 2023). The HH polarization and IW swath mode are chosen due to the comprehensive spatial coverage of Greenland. The satellites have near-polar, sun-synchronous orbit with a 12-day repeat

cycle and 175 orbits per cycle for each satellite. S1A and S1B have the same orbit plane with a 180° orbital phasing difference which results in an actual repeat cycle of 6 days with both satellites operating (Sentinel-1 SAR User Guide, 2023). For determining the timing of lake ice break-up, it is crucial to utilize the highest possible temporal resolution of the SAR data, which is why we use data from both ascending and descending orbits. The combination of both orbital modes and the high overlap of the acquisitions due to converging orbits close to polar regions leads to a coverage with a revisit frequency of below

2 days for most of Greenland (Sentinel-1 SAR User, Guide 2023). The prerequisite of both satellites being operational to ensure this revisit frequency constrains our study period to 2017-2021.

We use the Earth Engine Code Editor (2023) and load SAR data from the Earth Engine Data Catalogue (2023) which allows for performing an online analysis of large datasets with virtually no computational cost on a desktop computer. Accessing the data as Level-1 GRD product means that border noise removal, thermal noise removal, radiometric calibration and terrain

correction have already been performed following The Sentinel-1 Toolbox (2023) pre-processing steps leading to a calibrated and ortho-rectified product with a pixel size of 10 * 10 m. The ground range detection process projects the slant range coordinates of the radar data represented by range and azimuth onto the ellipsoid of the Earth resulting in a product which has approx. square spatial resolution and square pixel spacing (Sentinel-1 SAR Technical Guide, 2023). Border noise removal deals with low intensity noise and invalid data on scene edges. The thermal noise correction removes additive noise in sub-

swaths to help reduce discontinuities between sub-swaths for scenes in multi-swath acquisition modes (Sentinel-1 Algorithms, 2023). Radiometric calibration ensures that the intensity value represents the value of the reflectivity, i.e., the radar cross

section normalized to area (Moreira et al., 2013). This backscatter coefficient $\sigma_0$ can vary by several orders of magnitude and is therefore converted to decibel, as shown in Eq. (1).

$$\sigma_0 = 10 \, \log_{10} \sigma_{0 \, raw} \tag{1}$$

It measures whether the radiated terrain scatters the incident microwave radiation preferentially away from the SAR sensor ($\sigma_0 < 0$) or towards the SAR sensor ($\sigma_0 > 0$) (Sentinel-1 Algorithms, 2023). Terrain correction ensures that the location of any pixel in the SAR image is directly associated to the position on the ground. Radar only measures the projection of a three-dimensional scene on the radar coordinates slant-range and azimuth. This causes effects such as shadow for areas hidden from the radar illumination as well as foreshortening and layover manifested by a stretch and compression of sloped terrain (Moreira

et al., 2013). The terrain correction (or ortho-rectification) is based on the ASTER Global Digital Elevation Model (GDEM) (U.S./Japan ASTER Science Team, 2023) given the high-latitude location of the study area (> 60° N) (Sentinel-1 Algorithms, 2023).

## 2.3 Incoming Shortwave Radiation and Air Temperature

We use the current operational version of the Regional Atmospheric Climate Model (RACMO2.3p2), which is a high-
resolution regional climate model (RCM) adapted for high latitudes. It shows to capture spatial and temporal variability and absolute values well (Noël et al., 2019). Air temperature and shortwave radiation data from RACMO2.3p2 is used to bring lake-specific climatological variables into context with lake ice break-up timing. We utilize model outputs as opposed to measurements since field observations do not possess the spatial coverage for our large-scale study. The current operational version RACMO2.3p2 is validated against 37 automated weather stations (AWSs) on the GIS and proves to realistically

represent near-surface temperature ($0.73 < R^2 < 0.98$) and cloud conditions through shortwave and longwave radiation components ($0.85 < R^2 < 0.96$). This translates to biases in daily mean air temperatures at 2 m and incoming shortwave radiation of 0.14 °C and 4.8 W m$^{-2}$, the latter corresponding to a bias of 2.7 % (Noël et al., 2019).

For estimating excess radiation and energy due to variability in the break-up timing and investigating potential correlations with air temperature, we acquire lake-specific incoming shortwave radiation at the surface and air temperature at 2 m data as
climatological daily mean values for the period 1991-2020 from RACMO2.3p2 (Noël et al., 2019. The four nearest grid points of the model from the coordinates of each respective lake center point are selected in order to approximate the radiation and temperature data using Delaunay triangulation (Delaunay, 1934) with cubic interpolation.

## 3 Methods

### 3.1 Pre-Processing Lake Inventory and SAR Data

We retrieve SAR backscatter data of 14336 lakes which have a surface area $\geq 0.1$ km$^2$ to exclude potential inaccuracies due to the lake size. Time series of lakes with a temporal resolution below 2 days are excluded from the analysis to produce robust

results. While the acquisition frequency varies spatially and might be as high as 1 day, we assume a maximum temporal resolution of 2 days for the entire dataset which falls short of meeting the daily acquisition criteria of the GCOS ECV lake ice cover for climate monitoring (WMO et al., 2023b). Backscatter data which lacks a pronounced annual evolution and exhibits

strong uniformal characteristics are also excluded. This means that only lakes with a difference of $\geq 8$ dB in mean values of $\sigma_0$ between January/February (i.e., most certainly ice covered) and August/September (i.e., most certainly ice free) are considered.

## 3.2 Detecting Lake Ice Break-up from SAR Backscatter

The term "lake ice break-up" used in this study describes the timing, i.e., day of year (DOY), when at least 80 % of the lake

surface is liquid water and is therefore an approximation to the timing of "water clear of ice" (WCI) (WMO et al., 2023b). Once snowmelt starts on lake ice, water collects around the margins where it warms as it absorbs solar radiation and accelerates melting due to positive feedback (Jeffries et al., 2012). We assume that lake ice is longest present in the central areas of the lake and therefore aim to detect the presence or absence of ice in the central 20 % of the lake surface area which means that the $\sigma_0$ values are averaged for this central portion. This results in an area of approximately 0.02 km$^2$ for the smallest lake,

which corresponds to at least 200 pixels considered for averaging and proves to be a robust measure to identify the phenological state of lake ice.

**We apply a locally weighted scatterplot smoothing (LOWESS) filter to attenuate the temporal variability of $\sigma_0$ caused by varying incidence angles due to both ascending and descending orbits to ensure a more robust and confident ice break-up detection (Fig. 1a). We found the choice of 1 % of the data for LOWESS filtering to provide robust results for the analysis.**
**For each lake, a dynamic numerical threshold is applied in each year to identify the timing of ice break-up. This yearly threshold amounts to 25 % of the $\sigma_0$ difference between the 98$^{th}$ and 2$^{nd}$ percentile and must be at least 2.5 dB. The timing of lake ice break-up is detected when the absolute value of $\sigma_0$ decrease exceeds the threshold value within three consecutive acquisitions. Once the break-up timing is identified from the LOWESS data, the DOY of lake ice break-up is assigned to the lowest $\sigma_0$ value of the raw backscatter signal within five acquisitions ahead of the detected drop in the LOWESS data (Fig. 1a). The**
**detection algorithm is applied for the period starting from 1 May to exclude early misdetections.**

Figure 1a shows a typical SAR backscatter evolution for the period 2017 to 2021 with detected break-up timings. High $\sigma_0$ values (e.g., Nov-May) are governed by surface dry conditions of snow and ice (Unterschultz et al., 2009). The first major declines of $\sigma_0$ in a given year (e.g., May-Jun) indicates the onset of melt processes at the surface of the lake ice cover. A smooth, wet ice surface decreases the amount of backscatter due to specular reflection of the radar beam in a direction away

from the sensor. A rough, wet ice surface increases the amount of backscatter due to diffuse scattering, reflecting the radar beam nearly uniformly in all directions and directing a proportion of the incident energy back toward the sensor (Unterschultz et al., 2009). The progressing melt on the lake surface leading to a rougher, wetter surface explains the $\sigma_0$ recovery before the major backscatter decline in summer indicating lake ice break-up. We visually inspected timeseries of Sentinel-2 (S2) imagery

for selected lakes to support the observations of the backscatter dynamics and found that this is a typical $\sigma_0$ evolution during
the period of disintegration which is not related to refreezing processes.

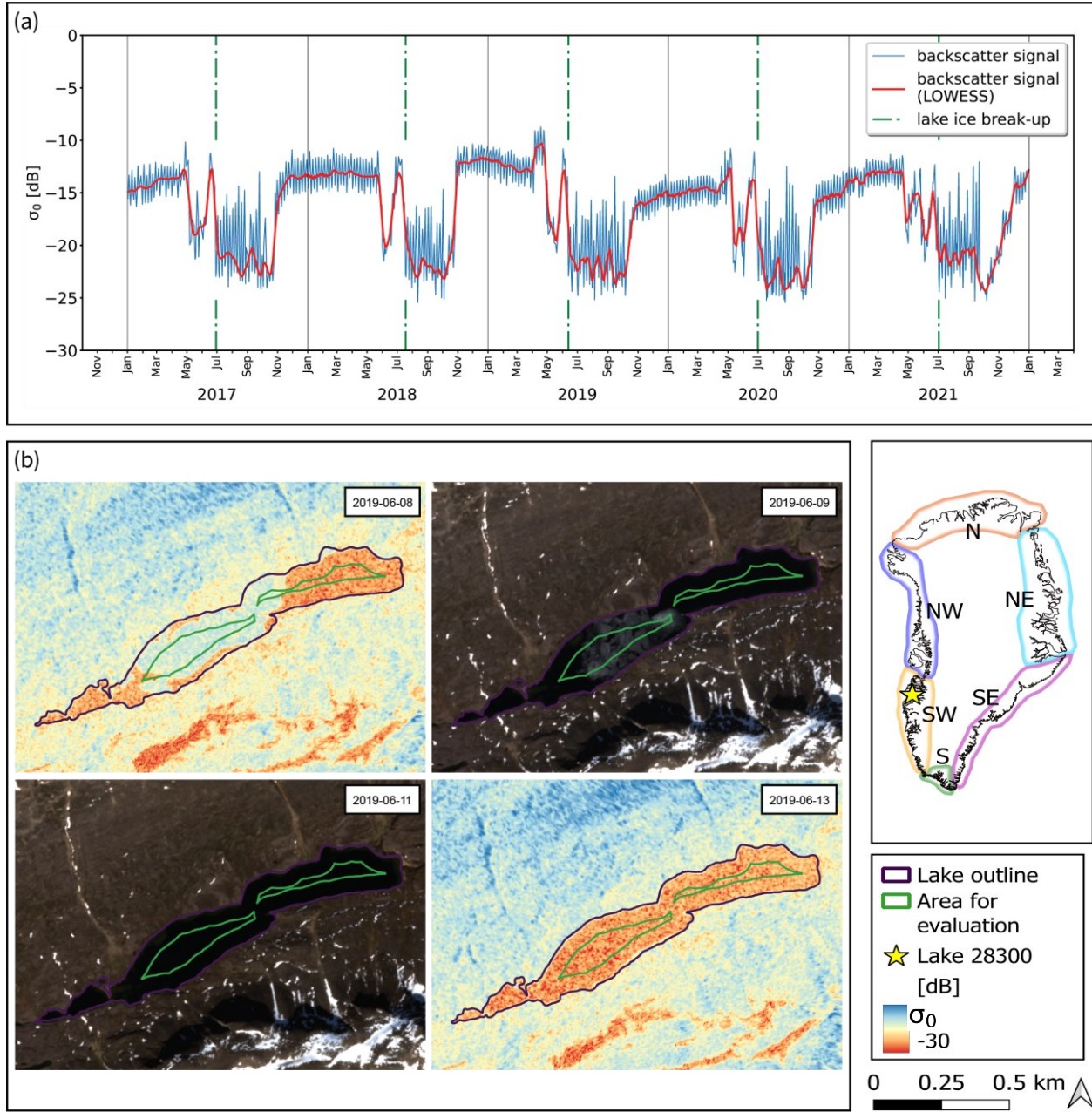

Figure 1: (a) Sentinel-1 (S1) synthetic aperture radar (SAR) backscatter ($\sigma_0$) and lake ice break-up timings for lake 28300 detected
from a dynamic numerical threshold assessing the $\sigma_0$ decline from being ice covered to open water. While the LOWESS smoothed
backscatter is utilized to confidently identify the period of break-up, the actual break-up timing is derived from the raw backscatter
signal. The $\sigma_0$ decline and recovery just before the apparent lake ice break-up indicates the onset of snow and ice melt. (b) S1 and
Sentinel-2 (S2) images of lake 28300 during break-up in 2019. The S2 scene shows water clear of ice (WCI) on 11 June 2019, while
the break-up timing from S1 is detected on 13 June 2019. (c) Regions of Greenland for the spatio-temporal analysis.

Figure 1a shows that in several years (e.g., from 2017 to 2020), the evolution of $\sigma_0$ might be clearly pronounced, while in other years (e.g., 2021) the backscatter decline and identification of the break-up timing is more complex. This is due to the nature of break-up processes being more complex due to melting on top and bottom (Jeffries et al., 2012) or varying acquisition conditions.

Figure 1b shows the results of lake 28300 in SW Greenland to demonstrate the detection of lake ice break-up in 2019 from SAR data compared to optical satellite imagery from the S2 mission. The detection algorithm identifies the lake ice break-up timing on 13 June 2019 from S1 backscatter data, while the S2 image shows water clear of ice on 11 June 2019 which can be taken as the accuracy of the method.

### 3.3 Analyzing Spatial Patterns of Lake Ice Break-Up Timing

The study area is divided into six regions (N, NE, SE, S, SW, NW) to explore spatio-temporal statistics (Fig. 1c). We chose a significance level of 0.05 to assess differences between regions and years, respectively, and use Pearson correlation coefficient (r) to assess relationships as linear correlations. Furthermore, we group lakes into sections of 1° N latitude and 100 m elevation, respectively, to assess spatial gradients and explore relationships between break-up timing and elevation, latitude as well as lake surface area. In the result statistics we include only lakes with detected ice break-up timings in every given year (2017-2021) to get robust detection statistics and to mitigate random detections. Furthermore, we manually remove obvious misdetections after visual inspection of the backscatter time series since we prioritize robust results statistics over a fully-automatically detected larger sample size that includes misdetections.

### 3.4 Assessing Climatological Variables in terms of Lake Ice Break-Up Timing

#### 3.4.1 Calculating Cumulative Positive Degree Days

In order to understand the relationship between the annual evolution of air temperature, incoming radiation and median lake ice break-up timings, we calculate the climatological mean of positive degree days (PDDs) from RACMO2.3p2 2 m daily air temperature averages for the period 1991 to 2020 and analyze them as cumulative values from 1 January until the median DOY of break-up. With this we support our discussion on the complexity regarding determining factors such as latitude, elevation, and radiation in context with cumulative PDDs.

#### 3.4.2 Calculating Excess Radiation and Energy

In order to determine the impact of varying lake ice cover on the radiation and hence the energy balance, we acknowledge several factors: (I) The surface albedo of the lake surface changes rather abruptly from ice to open water. That way, the same radiation and energy input gets converted to drastically higher energy amounts at the surface after ice break-up. (II) Depending at what time of the year the break-up happens, one will have a different impact on the energy balance. To illustrate the different impact a given change in lake ice break-up may have, we assume three arbitrary lakes of the same size at the same latitude

(and hence with the same arbitrary potential incoming shortwave radiation values) and show the impact in a conceptual way in Fig. 2. Assume that, for instance due to elevation differences, lake A typically breaks up some weeks before, B at the time of incoming radiation maximum, and C some weeks after. If we now assume changing conditions in a way that all three lakes break up 8 days (which corresponds to the median $MAD$ and is described in Eq. (4)) earlier than under 'regular' conditions, we see the following: For case A, less energy gets added, since it is at a time when radiation input is comparably low, while B changes the energy input more strongly. For C it means that despite the same temporal lag to the radiation maximum compared to A, the excess radiation and energy input gets higher since the period falls into a time with high incoming radiation. We coin the terms "excess radiation (input)" and "excess energy (input)" to describe the additional added radiation and energy due to an earlier lake ice break-up timing of 8 days compared to the observed median timings between 2017 and 2021 for each respective lake. (III) The lake size plays a role for total budget considerations: Large lakes have larger surface areas over which the energy can be accumulated. (IV) Finally, the shortwave radiation input is determined by latitude and altered by regional effects (e.g., cloud cover) or local effects (e.g., shading due to geometry) which is accounted for in the RACMO2.3p2 data (Noël et al., 2019).

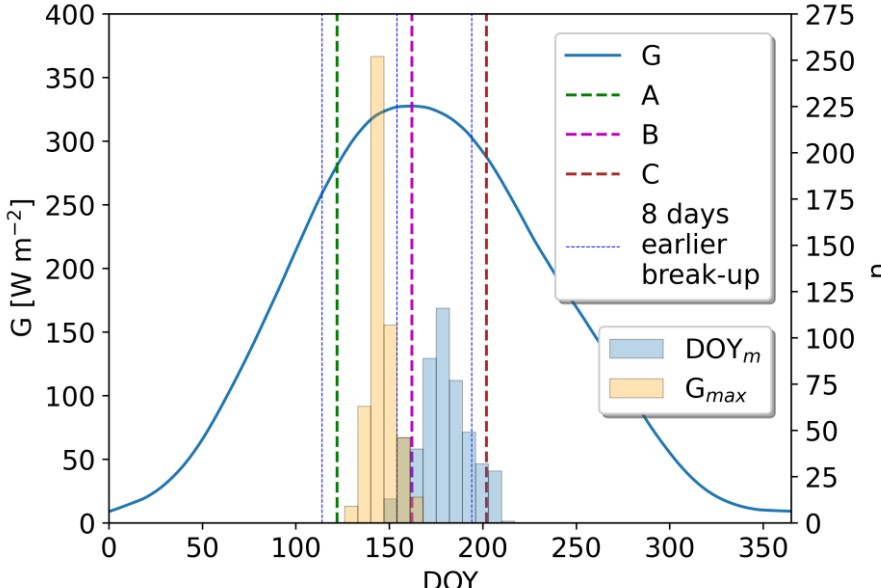

**Figure 2: Conceptual lake ice break-up timings of three arbitrary lakes (A, B, C), histograms of lake-specific median lake ice break-up $DOY_m$ and timing of maximum incoming radiation $G_{max}$. Assuming that all three lakes break up a fixed period earlier than their median break-up timings (in this case 8 days earlier which is the median variability in our data), it can be hypothesized that considering the annual evolution of global radiation $G$ at the surface, lake A receives less additional energy, lake B at the solar radiation maximum the most and lake C comparably higher additional energy than lake A. Median day of year (DOY) of $DOY_m$ and $G_{max}$ in our data are 178 and 145, respectively, indicating that most of the lakes will correspond to lake C.**

We quantify the excess radiation input $H_{SW}$ [J m$^{-2}$] for each lake as a consequence of an earlier break-up by integrating the shortwave radiation balance (incoming radiation $G$ [W m$^{-2}$] minus the reflected shortwave incoming radiation $R$ [W m$^{-2}$]) between 8 days before $DOY_m$ and the median lake ice break-up timing $DOY_m$, as shown in Eq. (2).

$$H_{SW} = \int_{DOY_m - MAD}^{DOY_m} G(t) - R(t) \, dt \tag{2}$$

The reflected shortwave incoming radiation $R$ [W m$^{-2}$] is calculated from the albedo difference $\Delta\alpha$ between the assumed values of snow-covered lake ice $\alpha_i$ (0.9) and open water $\alpha_w$ (0.1), as shown in Eq. (3).

$$R = G \, \Delta\alpha \qquad with \qquad \Delta\alpha = \alpha_i - \alpha_w = 0.8 \tag{3}$$

We hypothesize that an earlier lake ice break-up impacts the timing of the entire lake ice disintegration process from the melting of the initially dry snow cover on top of the lake ice until the melting of the bare ice surface itself. The short-term albedo development might be highly variable, impacting the transition from dry to wet snow, from wet snow to bare ice, and from bare ice to open water. We assume that the earlier break-up exhibits a shift of the entire disintegration period while its length is maintained, which means that the period of the snow-covered lake is 8 days shorter while the period of the lake having

an open water surface is 8 days longer. Therefore, the albedo difference $\Delta\alpha$ is expressed as the change from snow-covered lake ice $\alpha_i$ (0.9) to open water $\alpha_w$ (0.1) to quantify the excess radiation input $H_{SW}$.

The chosen 8 days for the hypothesized earlier lake ice break-up correspond to the median of the lake-specific median absolute deviation $MAD$ describing the variability of the annual break-up timings $DOY_i$ around the median $DOY_m$ of this period, as shown in Eq. (4),

$$MAD = median(|DOY_i - DOY_m|) \qquad with \qquad DOY_m = median(DOY_i) \tag{4}$$

where $DOY_i$ stands for the break-up timing in each respective year from 2017 to 2021. This can be regarded as a realistic period for an assumed earlier deviation of the break-up timings of all lakes to assess the impact of varying lake ice cover on the radiation balance.

Furthermore, we calculate the excess energy input $E_{SW}$ [J] for each lake by multiplying the excess radiation H$_{SW}$ [J m$^{-2}$] for

the 8-days-earlier lake ice break-up with the respective lake areas $A_i$ [m$^2$], as shown in Eq. (5).

$$E_{SW} = H_{SW} \, A_i \tag{5}$$

That way, we consider general radiation conditions, lake size and albedo change and express its reaction on a change in timing. Clearly, a change of the break-up timing at or just after the radiation maximum of a large lake will have a higher impact than a change later in the season for a small lake.

In order to make the results more tangible, we calculate what the excess energy inputs $E_{SW}$ mean in terms of mass and volume ice melt at the melting point ($m_i, V_i$) and water temperature increase ($m_w, V_w$). For this we convert the summed energy input $E_{SW}$ of all lakes using the latent heat of fusion $L_f$ (334000 J kg$^{-1}$), the specific heat capacity of water $c_w$ (4184 J kg$^{-1}$K$^{-1}$) and assumed densities of ice at the melting point $\rho_i$ (999 kg m$^{-3}$) and water close to freezing point $\rho_w$ (999 kg m$^{-3}$), as shown in Eq. (6) and Eq. (7).

$$m_i = \frac{\Sigma E_{SW}}{L_f} \qquad and \qquad V_i = \frac{m_i}{\rho_i} \tag{6}$$

$$m_w = \frac{\Sigma E_{SW}}{c_w} \qquad and \qquad V_w = \frac{m_w}{\rho_w} \qquad\qquad\qquad (7)$$

Our calculations show that lake surface area $A_i$ strongly determines the excess energy input $E_{SW}$ and explains more than 99 % of its variability in the dataset (Fig. D2). This allows for translating the excess energy input $E_{SW}$ to ice thickness melted or water depth warmed by 1 K across the respective lake surface area which are derived the from excess radiation input $H_{SW}$. For this estimate we ignore lake bathymetry and present the mean values of melted thickness of ice at the melting point $h_i$ [m] and depth of water $h_w$ [m] warmed by 1 K, as shown in Eq. (8) and Eq (9).

$$h_i = mean(\frac{H_{SW}}{L_f \, \rho_i}) \qquad\qquad\qquad\qquad (8)$$

$$h_w = mean(\frac{H_{SW}}{c_w \, \rho_w}) \qquad\qquad\qquad\qquad (9)$$

**3.5 Validating Detected Lake Ice Break-Up Timings**

The break-up detection is assessed and validated in three ways: (I) We utilize daily time-lapse images of three lakes (Badesø, Langesø, Quassi-sø) in vicinity of Kobbefjord (SW Greenland) between 2017 and 2020 (Abermann et al., 2019) to quantify the mean error of lake ice break-up of those lakes compared to the detection algorithm. (II) We use lake ice break-up data from observations, thermistor data and satellite imagery in the Kangerlussuaq area (SW Greenland) between 2017 and 2021 (Saros et al., 2019). While there are no corresponding lakes from the validation data included in our study due to a lack of pronounced radiometric properties, we compare median lake ice break-up timings between 11 lakes used for validation and 14 lakes in our study in vicinity to each other. (III) We access daily data from ESA CCI (Climate Change Initiative Lakes, 2023) for "lake ice cover (LIC)" to validate the break-up timing of two lakes (SW Greenland). This data is generated from MODIS imagery from both the Terra and Aqua satellite missions.

**4 Results**

**4.1 Lakes and Regions Suitable for Lake Ice Break-Up Detection and Analysis**

We restrict our analyses to the regions S, SW and NW since a comprehensive analysis for lakes in N, NE and SE is not possible due to challenging radiometric characteristics and/or temporal resolution (Table A1, Table A2), leaving a too small sample size. Since we only consider lakes with detected break-up timings in every given year between 2017 and 2021 and remove obvious outliers manually, we end up analyzing 563 lakes, which are 21 lakes in S, 450 lakes in SW and 92 lakes in NW. This corresponds to 0.4 % of all lakes or 6.8 % of the overall lake area in the inventory. The data coverage of RACMO2.3p2, however, allows for analyzing 491 lakes, which are 21 lakes in S, 406 lakes in SW and 64 lakes in NW regarding excess radiation and energy inputs and cumulative PDDs. This represents 0.3 % of all lakes or 6.8 % of the overall lake area in the inventory (Table A1, Table A2).

**4.2 Lake Ice Break-Up Detection Validation**

The detection of the lake ice break-up timings from SAR data proves to be conservative (i.e., later) compared to the lakes from all three validation approaches and allows characterizing break-up timing with a mean error of maximum 5 days. Yearly mean errors of the three compared lakes validated from time-lapse cameras range from 1-18 days exhibiting an overall mean error of 5 days (Table A3). Yearly differences between the median break-up timings of 14 lakes compared to 11 surrounding lakes used for validation in the Kangerlussuaq area range from 3-7 days with an overall difference of 5 days, indicating that the

interannual variability is well captured (Table A4). The two lakes validated from the ESA CCI data yield yearly mean errors ranging from 0-5 days with an overall mean error of 2 days (Table A5).

**4.3 Lake Ice Break-up Timing across S, SW, NW Greenland and Elevation Gradients**

Median break-up DOYs of all lakes range between 159 in 2019 and 191 in 2018, which corresponds to dates between 8 June and 10 July (Fig. 3). Regional annual median DOYs range from 168-212 (S), 159-191 (SW) and 153-188 (NW). Annual lake

ice break-up DOYs in S are significantly later for 2018 to 2021 compared to SW and NW. In 2017, the median break-up DOY in S exhibits no difference to SW and is significantly earlier compared to NW. The annual break-up timings in SW are significantly earlier in 2017 and significantly later in 2021 compared to NW.

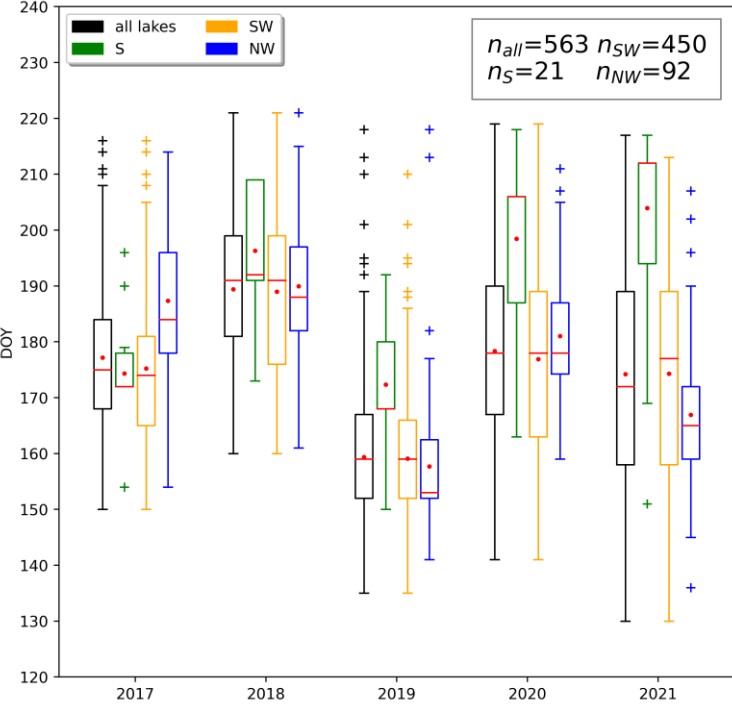

**Figure 3: Lake ice break-up timings of all studied lakes and grouped by region. Red lines indicate median values, while red dots**
**represent mean values. Median break-up timings are earliest in 2019 (8 June) and latest in 2018 (20 July). Lakes in the region S break up significantly later in 2018-2021 compared to the other regions, while lakes in NW tend to break up earliest. This can be attributed to mainly higher elevated lakes in S and mainly lakes close to sea level in NW.**

Lake-specific break-up timings as well as median break-up DOYs for 2017 to 2021 increase with elevation (Fig. 4a), while no confident latitudinal gradients nor correlations with lake surface area could be identified (Fig. 4b, Table C1). Median break-up DOYs for the period 2017 to 2021 increase by 3 DOY per 100 m elevation gain (r = 0.76, p < 0.01) (Fig. 4a), while yearly lake ice break-up DOYs show strong correlations ($0.51 \leq r \leq 0.78$, p < 0.01) with elevation exhibiting increases of 2-4 DOY per 100 m (Fig. C1). For a given elevation band, we find that lake ice in more northern latitudes tends to break-up later but exhibiting only weak correlations (Table C1).

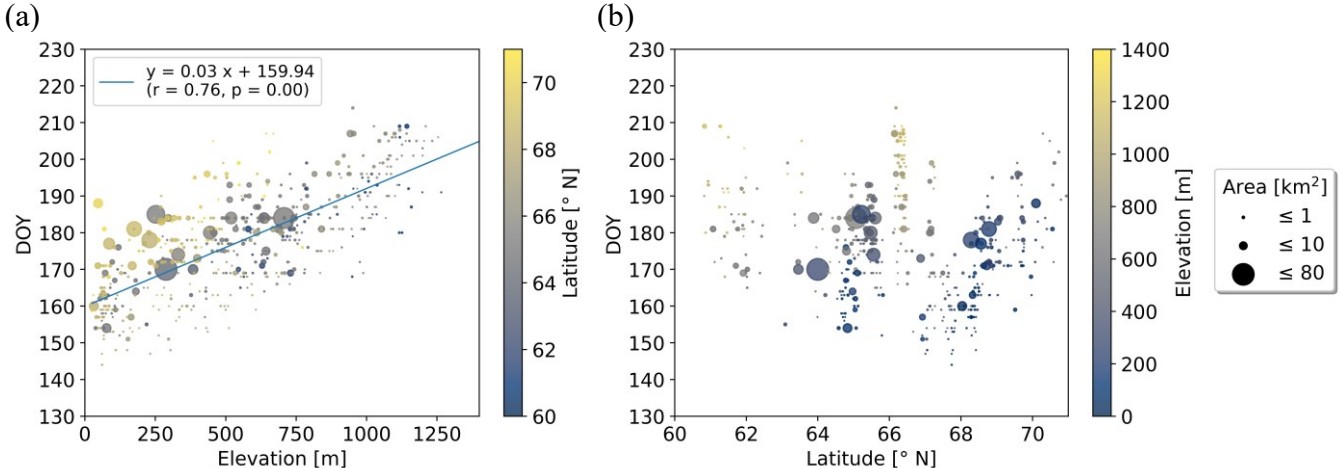

**Figure 4: Median break-up timings $DOY_m$ for the period 2017 to 2021 vs (a) elevation and (b) latitude. $DOY_m$ increase by 3 DOY per 100 m elevation gain exhibiting a strong correlation (r = 0.76, p < 0.01) while only a weak correlation with latitude can be identified.**

Subdivided into latitudinal bands of 1° between 60° N and 71° N, strong correlations (up to r = 0.89, p < 0.01) between break-up timing and elevation can be identified in several years. Those exhibit an increase of 3-6 DOY per 100 m depending on the latitudinal band as well as the elevation range and are significant in 43 out of the 55 yearly correlations (Fig. C2-Fig. C5). The median break-up dates for the period 2017 to 2021, except between 60-61° N and 70-71° N, show strong correlations ($0.64 \leq r \leq 0.85$, $p \leq 0.01$) increasing by 3-6 DOY per 100 m elevation increase.

### 4.4 Lake Ice Break-up Timing compared to cumulative PDDs until Lake Ice Break-Up

Figure 5 shows an increase of cumulative PDDs until lake ice break-up with increasing median break-up DOYs. A later break-up timing at lower latitudes can be observed in Fig. 5a when comparing lakes in a similar cumulative PDD range. Figure 5b shows that lakes with similar cumulative PDDs experience a later lake ice break-up at higher elevation. This is due to mainly higher elevated lakes at lower latitudes as opposed to lower elevated lakes at higher latitudes. Comparing two lakes at different elevation with a similar break-up timing, we see that a higher elevated lake with lower cumulative PDD values needs a comparably higher energy input (or less energy output) to accommodate for the same break-up timing as the lower elevated

lake with comparably higher cumulative PDDs. This is provided by a location at lower latitudes with comparably more incoming shortwave radiation.

(a)                                                                  (b)

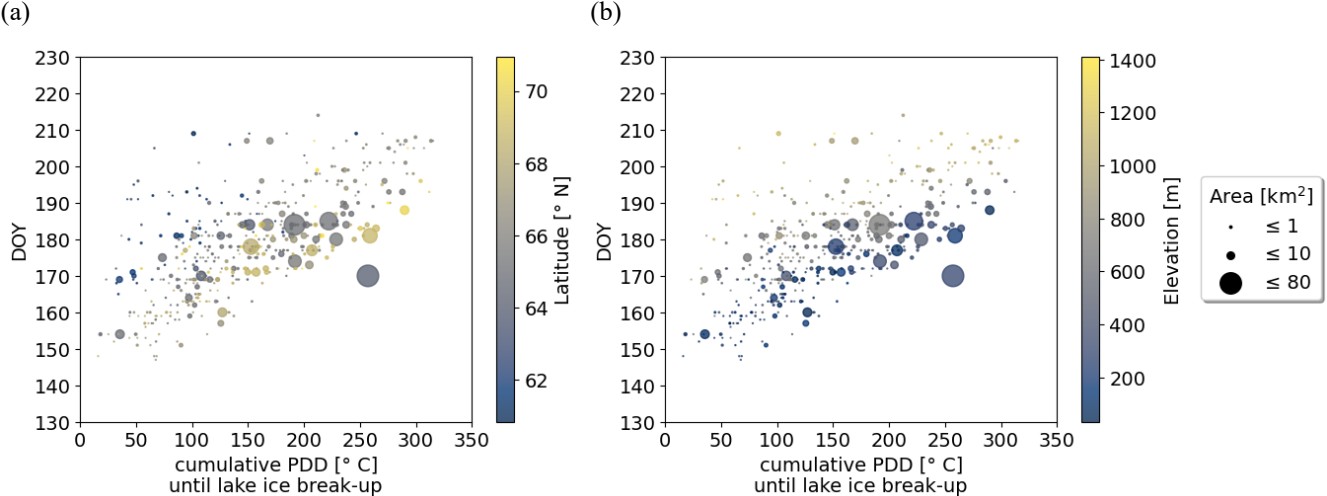

 **Figure 5: Median break-up timings $DOY_m$ for the period 2017 to 2021 vs cumulative positive degree days (PDD) until lake ice break-up with color signatures for (a) latitude and (b) elevation. The influence of air temperature and radiation on the break-up timing (i.e., in the energy budget) can be assessed when comparing two lakes at different elevation with a similar break-up timing. While the lower elevated lake has higher cumulative PDD values and is located at higher latitude, the higher elevated lake with lower cumulative PDDs is located further south.**

**4.5 Excess Radiation and Energy from earlier Lake Ice Break-Up**

Referring to the concept shown in Fig. 2, which describes the median lake ice break-up timings in relation to the timing of maximum incoming solar radiation, we find that virtually no lakes represent case A (< 0.01 %), approximately 5 % represent case B which are around the respective radiation maximum (± 8 days), while case C applies to approximately 95 %. The median time difference between the lake-specific maximum incoming radiation and median break-up amounts to 35 days

(Fig. D1). Figure D1 shows that excess radiation $H_{SW}$ is highest for lakes around and after the radiation maximum, while $H_{SW}$ values are decreasing with increasing later timing of the median break-up (i.e., increasing distance from the solar radiation maximum). Highest $H_{SW}$ values can be found at low latitudes as well as high-latitude lakes at lower elevation (Fig. 6). At lakes with similar latitude, higher $H_{SW}$ values are found at lower elevations, while at lakes with similar elevation, excess radiation values are typically higher at lower latitude.

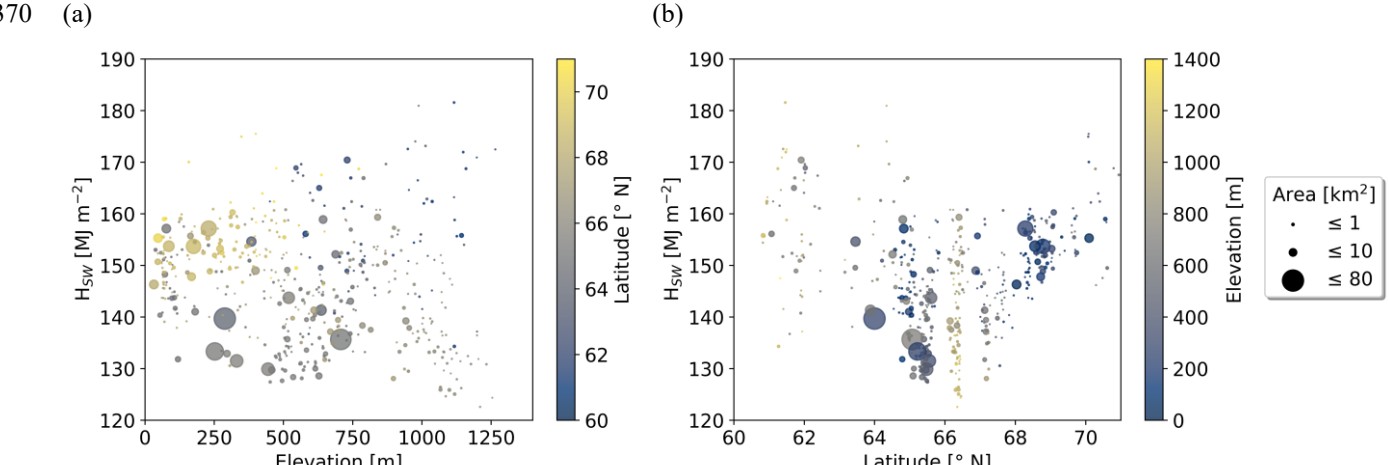

**Figure 6: Excess radiation $H_{SW}$ due to lake-specific break-up timings which are 8 days earlier compared to the median break-up timings for the period 2017 to 2021 vs (a) elevation and (b) latitude. High $H_{SW}$ values are found for lakes at low latitudes as well as at high latitudes with low elevation.**

Figure 7 shows the lake specific results for $E_{SW}$ as a function of latitude (Fig. 7a) and elevation (Fig. 7b). We clearly find larger $E_{SW}$ values for larger lakes and a weaker, yet visible dependence with latitude. Regarding a relation to elevation, we see generally larger $E_{SW}$ for lower elevations, which is partly due to the fact that larger lakes typically develop in lower elevations. Also, the fact that high-elevated lakes typically break up later after the radiation maximum plays a role in that regard. We find that the summed excess energy of all 491 analyzed lakes which amounts to 133250 TJ corresponds to melting 0.4 Gt ice or an ice cube of 7.4 km length. Likewise, the same energy input could heat up 31.9 Gt water or a water cube of 31.7 km length by 1 K.

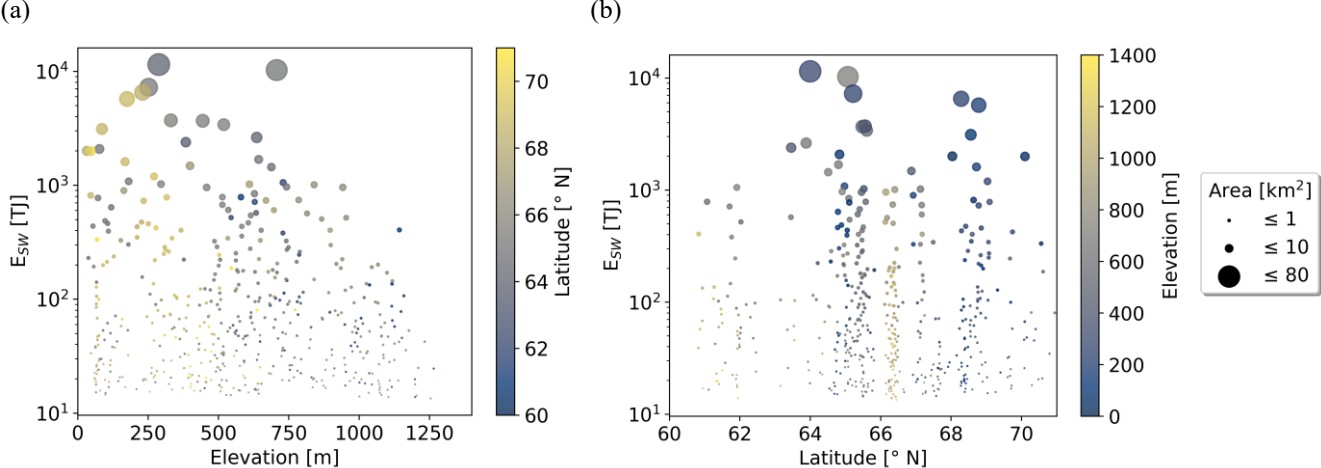

**Figure 7: Excess energy $E_{SW}$ due to lake-specific break-up timings which are 8 days earlier compared to the median break-up timings for the period 2017 to 2021 vs (a) elevation and (b) latitude. Lake surface area strongly determines the excess energy. Largest $E_{SW}$ are found at mid-latitudes and low elevation, since larger lakes typically developed there.**

Our calculations on excess energy show that lake surface area strongly determines the added energy and explains more than 99 % of its variability in the dataset (Fig. D2). Referring the added energy in terms of ice melt and water temperature rise to the lake-specific areas in a simplified way which ignores lake bathymetry, we find that the excess energy input averagely corresponds to melting $0.4 \pm 0.1$ m thick ice or heating up a water depth down to $35 \pm 3$ m by 1 K across the entire surface areas. If we upscale our results to all 100486 lakes in S, SW and NW (< 71° N) Greenland based on the strong relationship between excess energy and surface area while assuming similar radiation conditions and lake ice break-up variabilities, we estimate an additional energy input of $1.8 * 10^6$ TJ which corresponds to melting 5.8 Gt ice at the melting point or warming 432.3 Gt water by 1 K. This number of lakes corresponds to 64.5 % of all lakes or 62.1 % of the overall lake area in the inventory (Table A1, Table A2). To put this into perspective, the upscaled mass estimate of ice melt corresponds to approximately 30 to 60 % of the volume of Greenland peripheral glaciers published in recent studies (Hock et al., 2023).

## 5 Discussion

### 5.1 Limitations and Potentials

In our study, we apply a dynamic numerical threshold on the annual SAR backscatter evolution to establish an automated detection of lake ice break-up timing. Differences in the dielectric properties between water and ice which are manifested in a decline in radar backscatter (Unterschultz et al., 2009) and the fast transition from an ice-covered to an open water surface allow for an automated detection. In contrast, different modes of lake surface freeze-up as well as high local and inter-annual variability of backscatter during long periods of gradual freezing makes an automated detection of freeze-up very challenging. This limitation in the automated detection method constrains this study to an analysis of the variability of lake ice break-up timing as opposed to an analysis of spatial and temporal variations in the length of the period in which the lake surface is frozen. However, we argue, that regarding the impact of lake ice presence on surface energy balance, the break-up timing of lake ice is more relevant than the timing of freeze-up. In our data we see that freeze-up typically occurs in October or November when incoming solar radiation is lower than during the timing of break-up (yearly median break-up timings vary between 8 June and 10 July) (Fig 2).

Our presented vertical gradients of break-up timings in the order of a few days must be interpreted in regards with temporal limitation of the data and method. The SAR data with an acquisition resolution of 2 days implies a maximum accuracy of ± 1 day for detecting lake ice break-up, while the validation indicates that our automated detection exhibits a mean error of 2-5 days. The GCOS ECV requirements state that the temporal resolution of our study allows for contrasting extreme ice years, numerical weather forecasting and assessing lake models (WMO et al., 2023b). In a related study, Murfitt and Duguay (2020) also utilized S1 high-density time series data to monitor ice phenology of Lake Hazen located in Nunavut, Canada (71.05° W, 81.78° N). They found mean errors of 3-7 days for comparing sectional WCI dates and 3-5 days for a pixel-based ice-off comparison, which is in agreement and in the order of our validation results.

The estimates of the excess radiation and consequently excess energy are based on assuming a shift of the entire break-up process while having its length maintained. This means that the excess energy is assumed to be available for the open water surface of the lake and is not utilized during the break-up process. Short-time albedo development during the disintegration process of the lake surface might be highly variable and changes in the configurations and lengths of the transition from dry to wet snow, from wet snow to bare ice and from bare ice to open water might vary greatly with elevation, latitude and local climates. A site-specific characterization of albedo evolution during lake ice break-up might be a subject for further studies and improve local excess energy estimates.

## 5.2 Lake Ice Break-Up in context of Coastal Climate and Topography

We assess that elevation more strongly determines lake ice break-up timing in Greenland than latitude does. For both all lakes and lakes grouped by latitudinal sections, we demonstrated that there are strong correlations between median break-up timings and elevation as well as yearly break-up timings and elevation. The significantly later timing of yearly median break-up DOYs in S compared to SW and NW in several years can be explained by the hypsometry of the terrain and the distribution of lakes with elevation (fewer lakes close to sea level). Local topography and extent of fjord systems can have a strong influence on the timing of ice break-up which may be due to local climate influences and sea ice. This is demonstrated by lakes with early break-up timings in close vicinity to fjords such as between 67.5° N and 68.5° N (Fig. 8a: orange arrow) as opposed to lakes with late break-up timings at high elevation areas (> 1000 m) being surrounded by ice bodies such as between 66.0° N and 66.5° N (Fig. 8a: blue arrow). When discussing break-up timing in context of topography, the spatial distribution of snowfall must also be considered. The amount of snowfall may greatly influence the variability of lake ice due to its insulating properties on top of the lake ice surface as well its role during lake ice build-up and disintegration. The difference in break-up timing between coastal lakes which is earlier at 67.5-68.5° N compared to being later at > 68.5° N may be attributed to higher snow accumulation rates which can be as much as three times higher in the latter case, as shown by Bales et al., 2009.

Imrit and Sharma (2021) found that climate change, warmer local air temperatures, and teleconnection patterns are able to explain on average approx. 60 % of the variation in ice phenology of lakes in Northern America and Europe. On average, approx. 40% of the variation remained unexplained and could be attributed to local weather conditions, such as solar radiation inputs, wind, snow cover and lake and landscape characteristics, such as lake depth, elevation and fetch. This is in line with our interpretation of variability in the timing of lake ice break-up which may be greatly attributed to local weather, climate and landscape characteristics in coastal Greenland.

Lake ice break-up timings of the years 2018 (Fig. 8b) and 2019 (Fig. 8c) which exhibit the latest (DOY 191; 10 July) and earliest (DOY 159; 8 June) median lake ice break-up DOYs within the observed period are in line with temperature observations. Mean summer and July temperatures in 2019 were among the 6 warmest years (1981-2019), while comparably cooler JJA air temperatures at Greenland coastal stations were recorded in 2018 (Hanna et al., 2021).

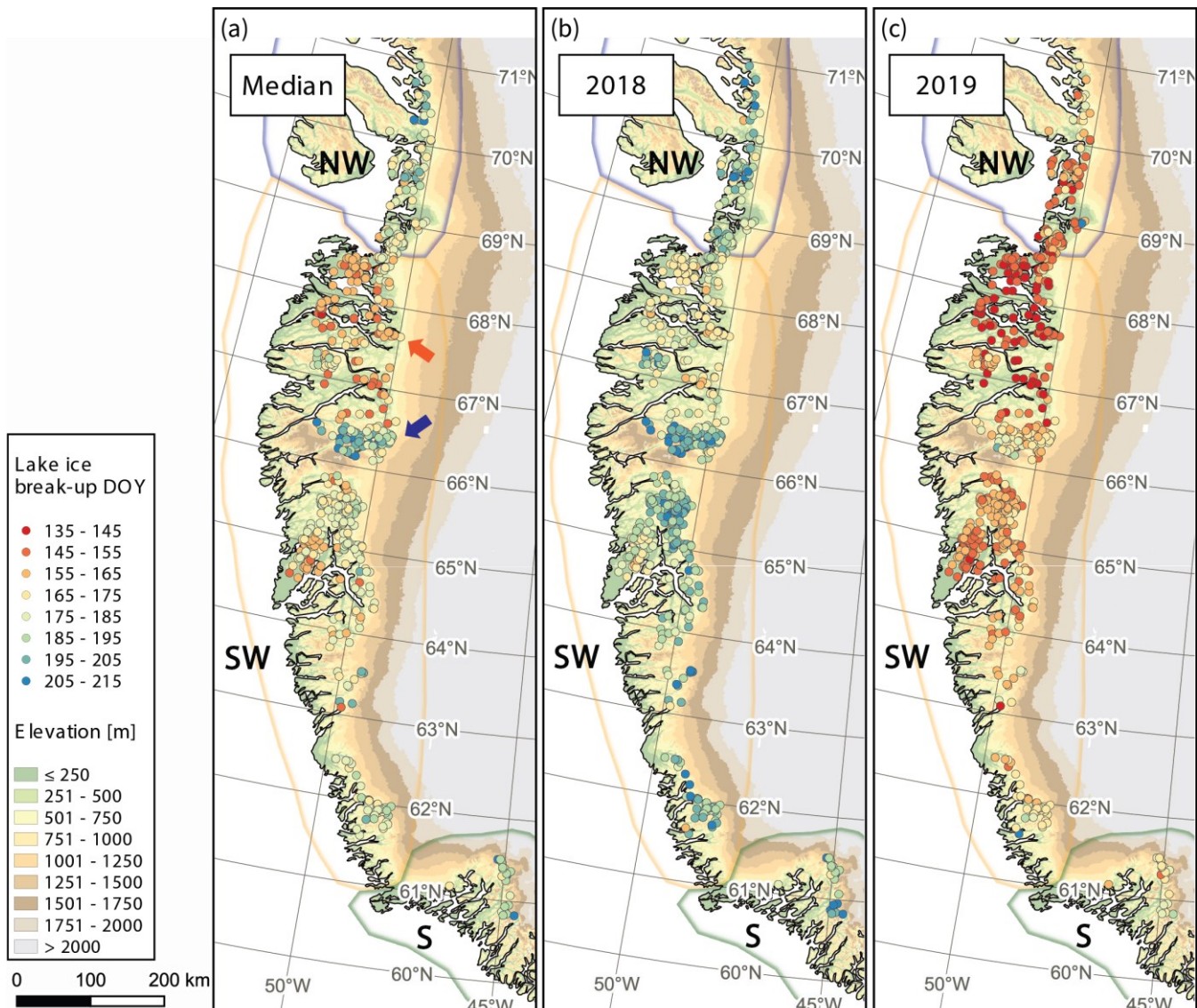

**Figure 8: (a)** Median lake-specific break-up timings for the period 2017 to 2021, and break-up timings for **(b)** 2018 and **(c)** 2019. Local topography such as higher elevation (blue arrow) and extent of fjord systems (orange arrow) can strongly influence lake ice break-up dates. The year 2018 exhibits the latest break-up timings while the year 2019 shows the earliest break-ups in our studied period, corresponding to median dates 10 July (DOY 191) and 8 June (DOY 159), respectively.

### 5.3 Lake Ice Break-Up in Greenland compared to other Study Sites

L'Abée-Lund et al. (2021) studied the phenology of 101 Norwegian lakes between 1890-2020 covering a latitudinal range of 58.2-69.9° N and found stronger correlations between the average timing of ice break-up and elevation (3.4 days per 100 m, r = 0.63, p < 0.01) when compared to the correlations between break-up timing with latitude (2.3 days per 1° N, r = 0.35, p < 0.01.). Williams et al. (2004) and Williams and Stefan (2006) assessed lake ice break-up timing of approximately 140

lakes in North America (between 40.0° N and 82.5° N) from records between 1848 and 1997 and found that there is a strong
relationship between break-up timing and latitude, arguing that geographic latitude is a good indicator of climate. They showed
that only a weak relationship between elevation and the timing of break-up was observed when grouping the data by region,
presenting an increase of break-up timing of 2 days per 100 m elevation increase. Zhang et al. (2021) found a strong correlation
between latitude and lake ice-break up timing ($R^2$ = 0.75) and only a weak relationship with elevation for 4241 lakes in Alaska
over the period 2000-2019.

In our study which covers lakes between 60° N and 71° N, however, we find strong correlations between break-up timing and
elevation and only a weak relationship with latitude for lakes at similar elevation. This again highlights the influence of the
spatial configuration of lakes in Greenland determined by the proximity to both the ocean and the GIS, the presence of fjord
systems and the steep slopes.

**5.4 Lake Ice Break-Up and Climate Change**

Magnusson et al. (2000) found that lake ice break-up dates in the Northern Hemisphere from 1846 to 1995 became on average
6.5 days earlier per century, corresponding to an air temperature increase of approximately 1.2 °C during this period. They
claim that the interannual variability of break-up timing increased since 1950. Hallerbäck et al. (2022) studied lake ice
phenology in Sweden from observations spanning the period 1700-2014 and found that the break-up timing of northern lakes
(> 60° N) advances by 4.4 days per 1 °C of warming. Our observed variability of ± 8 days based on the period 2017-2021 lies
above the observed break-up shift over 100 years, which might be either attributed to the increased variability in recent years,
or due to the exceptional years 2018 and 2019 greatly skewing the variability in the short observational period. Imrit and
Sharma (2021) showed that 18 lakes around North America and Europe were thawing 9 days earlier per century over the past
156-204 years. They argued that by adding 24 years of data to the previous study of Magnusson et al. (2000), rates of lake ice
loss are almost 1.5 times faster which may be attributed to warmer air temperatures and a higher prevalence of extreme events
in recent decades.

Increased rainfall and especially a prolonged rainfall period may have indirect effects on the lake ice break-up timing due to
snowpack heating through percolation and refreeze and a subsequent melt-albedo feedback initiated by heat and rainfall (Box
et al., 2022; Box et al., 2023). Projections of temperature increase in the magnitude of 1.5-5.0° C until 2100 and a trend of
precipitation change from snowfall to rainfall (IPCC, 2023) may result in an earlier break-up timing of lakes in Greenland and
an increased variability until the end of this century. Using a one-dimensional thermodynamic model for simulating lake ice
conditions for 471 lakes in the Northern hemisphere, Huang et al. (2022) estimated that the global mean lake-area averaged
ice break-up date is projected to advance by 20 ± 7 days over 2020-2100, which translates to an average ice duration decrease
by 9.9 days for a 1 °C warming. They claim that the lake ice break-up timing in the Canadian Arctic, northern Siberia and
close to the Barents Sea can be clearly linked to the enhanced polar warming over sea-ice areas in the Arctic Ocean in the cold
season. They hypothesize that the opening of sea ice in summertime leads to the absorption of anomalous heat by the ocean.
In wintertime, the excess heat can reduce sea-ice coverage which in turn generates large heat fluxes from the warmer ocean to

the highly stratified winter atmosphere. The atmospheric heating spreads further to neighboring land areas, where it can influence lake ice. Furthermore, Huang et al. (2022) discussed the ice-albedo feedback on lake ice. Due to future ice loss and reduced surface albedo, lakes will absorb more shortwave radiation in the extended ice-free season. The excess heat can reduce

ice cover in winter, therefore leading to earlier ice break-up, which in turn triggers a positive feedback in spring.

**5.5 Implications of Lake Ice Break-Up Variability**

The magnitudes of our estimated excess energies from earlier break-up timings indicate potentially vast changes in the energy balance with increasingly earlier break-up dates and increasing variabilities due to a changing climate. Furthermore, changes in lake ice phenology which influence sub-surface mixing, temperature and light conditions have a direct impact on the

ecosystem. These ramifications due to an earlier lake ice break-up timing may manifest themselves in increases in hypolimnetic oxygen concentrations because of changes in the lake mixing regime, potential earlier and more $CO_2$ emissions by lakes during lake ice melting, earlier thermal stratification, a longer open-water season for warming, higher water temperatures, increased evaporation rates and potentially decreasing water levels. Furthermore, shifts in phytoplankton biomass, food web dynamics and community composition may occur due to unpredictable changes in lake mixing regimes (Imrit and Sharma, 2021). Below-

ice aquatic biodiversity is often tied to the presence of ice, which in turn impacts the seasonal cycle of prey and predators, nutrient cycling, dissolved oxygen and the timing of algal blooms which may impact ecological processes even in summertime (Huang et al., 2022). Lake ice break-up timing may also have further downstream implications, such as freshwater input into fjords in regard to transport of nutrients and sediments which influence the geochemical composition and ultimately the marine primary production (Abermann et al., 2021).

Besides these direct impacts in natural systems, the variability in lake ice break-up exhibits several anthropogenic implications. Climate change is likely influencing the thickness of lake ice as well as the timing of breakup, which has a significant impact on both reservoir inflows and outflows for hydropower (Cherry et al., 2017) as well as on its infrastructure such as damns, spillways, channels, reservoirs, tunnels, inlets, and outlets (Gebre et al., 2013). Changes in phenological ice regimes will make access to lakes more uncertain and potentially hazardous and may impact traditional subsistence-based lifestyles which are

dependent on the natural network for access to isolated communities, remote industrial developments, hunting, fishing, herding and trapping areas (Prowse et al., 2011).

**6 Conclusion**

We demonstrated that temporal high-resolution S1 SAR data can be utilized to detect lake ice break-up timings in SE, S, SW and NW Greenland. Our presented lake ice break-up timing results prove to be robust and conservative (i.e., later) with a mean

error of maximum 5 days and allow for a spatio-temporal characterization. We show that median lake ice break-up timings for the period 2017 to 2021 increase by 3 DOY per 100 m elevation increase, while no strong correlations can be found regarding latitude or lake area.

The 491 studied lakes exhibit a typical variability in break-up timing of ± 8 days. When we assume the break-up timing being 8 days earlier for each lake, the introduced excess energy corresponds to melting 0.5 m thick ice at the melting point or heating up a water depth down to 35 m by 1 K across the respective surface areas. Scaling up our results to 100486 lakes across S, SW and NW Greenland, the excess energy input amounts to approximately $1.8 * 10^6$ TJ for the hypothesized earlier lake ice break-up. The variability in lake ice break-up timing of the studied lakes in Greenland for 2017-2021 is above the observed variability of related lake ice phenology studies in other regions over the last century which may be attributed to exceptional temperature years in recent years and the comparable short study period. However, with progressing climate change an increased interannual variability and earlier timing in lake ice break-up can be expected.

Excluding data from days with high wind speeds or coupling the SAR-based detection with optical detections from satellite systems (e.g., S2) might yield more robust results with a higher accuracy but might also additionally decrease the temporal resolution. Applying machine learning or deep learning algorithms as a next step might further improve the break-up detection and increase sample size. There is potential for exploring the relationship between break-up timing and climatological variables and assess the impact on the energy budget in greater detail by incorporating a variety of parameters into more complex models. Coupling satellite-derived break-up results with a greater number of ground observations and in-situ measurements of meteorological variables might further improve remotely sensed break-up detections. A continued study of lake ice break-up timings detected from spatial and temporal high-resolution SAR data such as from S1 will be of high importance in terms of climate monitoring with more data available from increasing operational periods. We aim at applying our algorithm for an analysis of lake ice break-up timing on a global scale.

## Data availability

https://doi.org/10.5281/zenodo.10577480

## Author contribution

Christoph developed the detection algorithm, processed the data, produced all results and plots, and drafted the manuscript. Jakob developed the idea and greatly determined the scope and direction of the study and displayed results. Tiago was involved and contributed the RACMO2.3p2 data to the study.

## Competing interests

The authors declare that they have no conflict of interest.

## Acknowledgements

We thank Václava Hazuková and Jasmine Saros for providing observations on lake ice break-up dates for lakes in the Kangerlussuaq area (SW Greenland). Their data greatly contributed to validating our detection algorithm.

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

**Appendix A**

Table A1: Number of lakes through the different analysis steps in our study as well as their relative coverage compared to the entire lake inventory. We found that only lakes in S, SW and NW Greenland are suitable to perform a spatio-temporal analysis of break-up timings and a statistical analysis considering climatological data. [1] lakes with suitable temporal and radiometric characteristics for the automated lake ice break-up detection algorithm, [2] lakes used for spatio-temporal analysis, [3] lakes used for analysis of excess radiation, energy and cumulative PDDs, [4] lakes used for upscaled results of excess energy.

| | | inventory | A ≥ 0.1 km² | after pre-processing [1] | with all DOYs detected | after outlier removal[2] | with climate data[3] | upscaled excess energy[4] |
|---|---|---|---|---|---|---|---|---|
| all lakes | | 155870 | 14336 | 1693 | 828 | 563 | 491 | 100486 |
| N | | 18631 | 1613 | 4 | 3 | 0 | 0 | 0 |
| NE | | 19363 | 1973 | 1 | 1 | 0 | 0 | 0 |
| SE | abs. | 8402 | 547 | 2 | 0 | 0 | 0 | 0 |
| S | | 11301 | 792 | 54 | 33 | 21 | 21 | 11301 |
| SW | | 79147 | 7667 | 1360 | 640 | 450 | 406 | 79147 |
| NW | | 19013 | 1741 | 272 | 151 | 92 | 64 | 10024 |
| | | | | | | | | |
| all lakes | | 100.0% | 10.0 % | 1.1 % | 0.5 % | 0.4 % | 0.3 % | 64.5 % |
| N | | 12.0% | 1.0 % | < 0.1 % | < 0.1 % | 0.0 % | 0.0 % | 0.0 % |
| NE | | 12.4% | 1.3 % | < 0.1 % | < 0.1 % | 0.0 % | 0.0 % | 0.0 % |
| SE | rel. | 5.4% | 0.4 % | < 0.1 % | 0.0 % | 0.0 % | 0.0 % | 0.0 % |
| S | | 7.3% | 0.5 % | < 0.1 % | < 0.1 % | < 0.1 % | < 0.1 % | 7.3 % |
| SW | | 50.8% | 4.9 % | 0.9 % | 0.4 % | 0.3 % | 0.3 % | 50.8 % |
| NW | | 12.2% | 1.1 % | 0.2 % | 0.1 % | 0.1 % | 0.0 % | 6.4 % |

**Table A2: Surface area [km²] of lakes through the different analysis steps in our study as well as their relative coverage compared to the entire lake inventory. We found that only lakes in S, SW and NW Greenland are suitable to perform a spatio-temporal analysis of break-up timings and a statistical analysis considering climatological data. [1] lakes with suitable temporal and radiometric characteristics for the automated lake ice break-up detection algorithm, [2] lakes used for spatio-temporal analysis, [3] lakes used for analysis of excess radiation, energy and cumulative PDDs, [4] lakes used for upscaled results of excess energy.**

| | | inventory | A ≥ 0.1 km² | after pre-processing [1] | with all DOYs detected | after outlier removal [2] | with climate data [3] | upscaled excess energy [4] |
|---|---|---|---|---|---|---|---|---|
| all lakes | | 14183 | 11879 | 2770 | 1485 | 971 | 925 | 8806 |
| N | | 2165 | 1876 | 12 | 12 | 0 | 0 | 0 |
| NE | | 2198 | 1896 | 36 | 36 | 0 | 0 | 0 |
| SE | abs. | 372 | 260 | 11 | 0 | 0 | 0 | 0 |
| S | | 570 | 416 | 29 | 18 | 15 | 15 | 570 |
| SW | | 7381 | 6214 | 2310 | 1148 | 810 | 788 | 7380 |
| NW | | 1497 | 1217 | 372 | 271 | 146 | 122 | 854 |
| | | | | | | | | |
| all lakes | | 100.0 % | 83.8 % | 19.5 % | 10.5 % | 6.8 % | 6.5 % | 62.1 % |
| N | | 15.3 % | 13.2 % | 0.1 % | 0.1 % | 0.0 % | 0.0 % | 0.0 % |
| NE | | 15.5 % | 13.4 % | 0.3 % | 0.3 % | 0.0 % | 0.0 % | 0.0 % |
| SE | rel. | 2.6 % | 1.8 % | 0.1 % | 0.0 % | 0.0 % | 0.0 % | 0.0 % |
| S | | 4.0 % | 2.9 % | 0.2 % | 0.1 % | 0.1 % | 0.1 % | 4.0 % |
| SW | | 52.0 % | 43.8 % | 16.3 % | 8.1 % | 5.7 % | 5.6 % | 52.0 % |
| NW | | 10.6 % | 8.6 % | 2.6 % | 1.9 % | 1.0 % | 0.9 % | 6.0 % |


**Appendix B**

**Table A3: Lake ice break-up validation from time-lapse cameras (Abermann et al., 2019) in SW Greenland.**

| year | Badesø | | | Langesø | | | Quassi-sø | | | yearly mean error | overall mean error |
|---|---|---|---|---|---|---|---|---|---|---|---|
| | Lake ID: 16515 13 m a.s.l. | | | Lake ID: 17830 21 m a.s.l | | | Lake ID: 17831 226 m a.s.l. | | | | |
| | DOY val. | DOY S1 | diff. | DOY val. | DOY S1 | diff. | DOY val. | DOY S1 | diff. | | |
| 2017 | 168 | 169 | 1 | 167 | 168 | 1 | 173 | 175 | 2 | 1 | 5 |
| 2018 | 171 | 189 | 18 | 171 | - | - | 184 | - | - | 18 | |
| 2019 | 149 | 152 | 3 | 148 | 152 | 4 | 159 | - | - | 4 | |
| 2020 | 161 | - | - | 166 | 171 | 5 | 179 | 185 | 6 | 6 | |

**Table A4: Lake ice break-up validation from observations, thermistor data and satellite imagery of 11 lakes (Saros et al., 2019) in the Kangerlussuaq area compared to 14 lakes from our study close by.**

| year | median DOY val. | median DOY S1 | diff. | overall mean difference |
|---|---|---|---|---|
| 2017 | 155 | 162 | 7 | 5 |
| 2018 | 164 | 169 | 5 | |
| 2019 | 139 | 144 | 5 | |
| 2020 | 150 | 154 | 4 | |
| 2021 | 150 | 153 | 3 | |

**Table A5: Lake ice break-up validation from ESA CCI data (Climate Change Initiative Lakes, 2023) in SW Greenland.**

| year | Lake ID: 70398 519 m a.s.l. | | | Lake ID: 70316 707 m a.s.l | | | yearly mean error | overall mean error |
|---|---|---|---|---|---|---|---|---|
| | DOY val. | DOY S1 | diff. | DOY val. | DOY S1 | diff. | | |
| 2017 | 177 | 181 | 4 | 172 | 175 | 3 | 4 | 2 |
| 2018 | 199 | 199 | 0 | 199 | 201 | 2 | 1 | |
| 2019 | 160 | 165 | 5 | 159 | 160 | 1 | 3 | |
| 2020 | 182 | 184 | 2 | 183 | 184 | 1 | 2 | |

## Appendix C

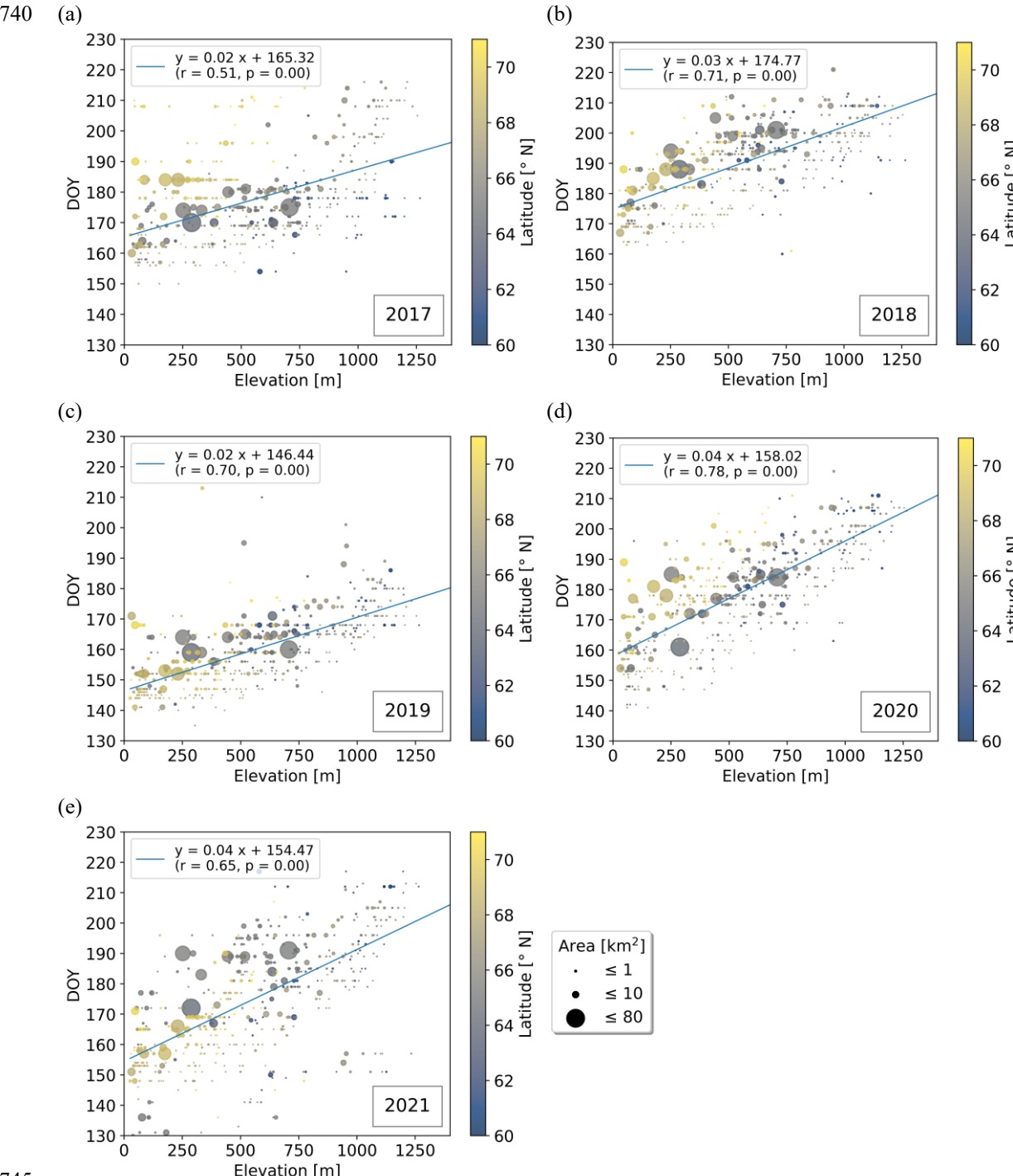

**Figure C1: Lake ice break-up timings vs elevation for the years (a) 2017 to (e) 2021. Break-up dates increase by 2-4 DOY per 100 m elevation gain exhibiting strong correlations (0.51 ≤ r ≤ 0.78, p < 0.01).**

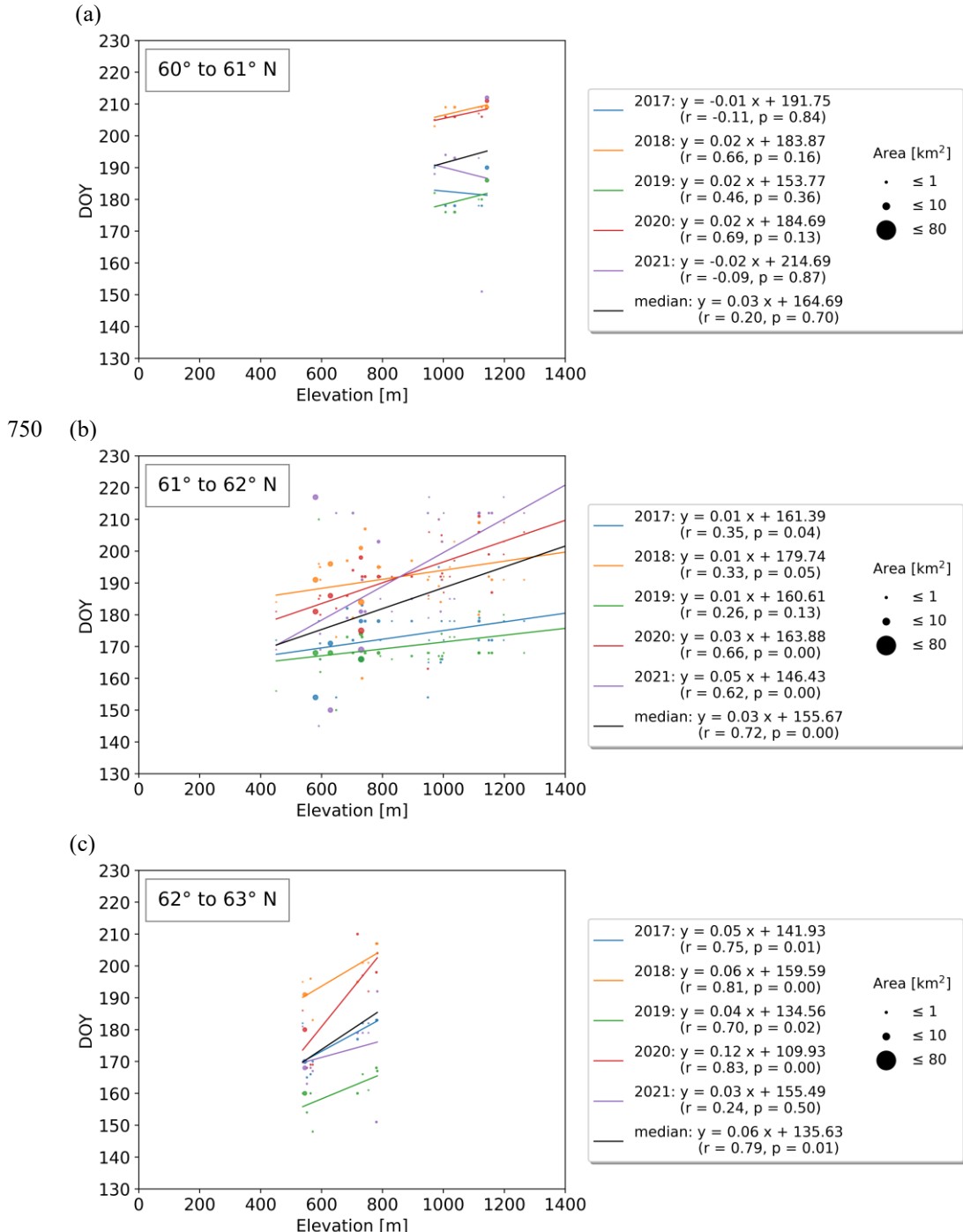

**Figure C2: (a)-(c) Yearly lake ice break-up timings vs elevation grouped by 1° latitude between 60° and 63° N. Several years exhibit strong correlations between break-up timing and elevation which increase by 3-6 days per 100 m elevation gain. Significance values of p = 0.00 in the figures indicate highly significant relations of p < 0.01.**

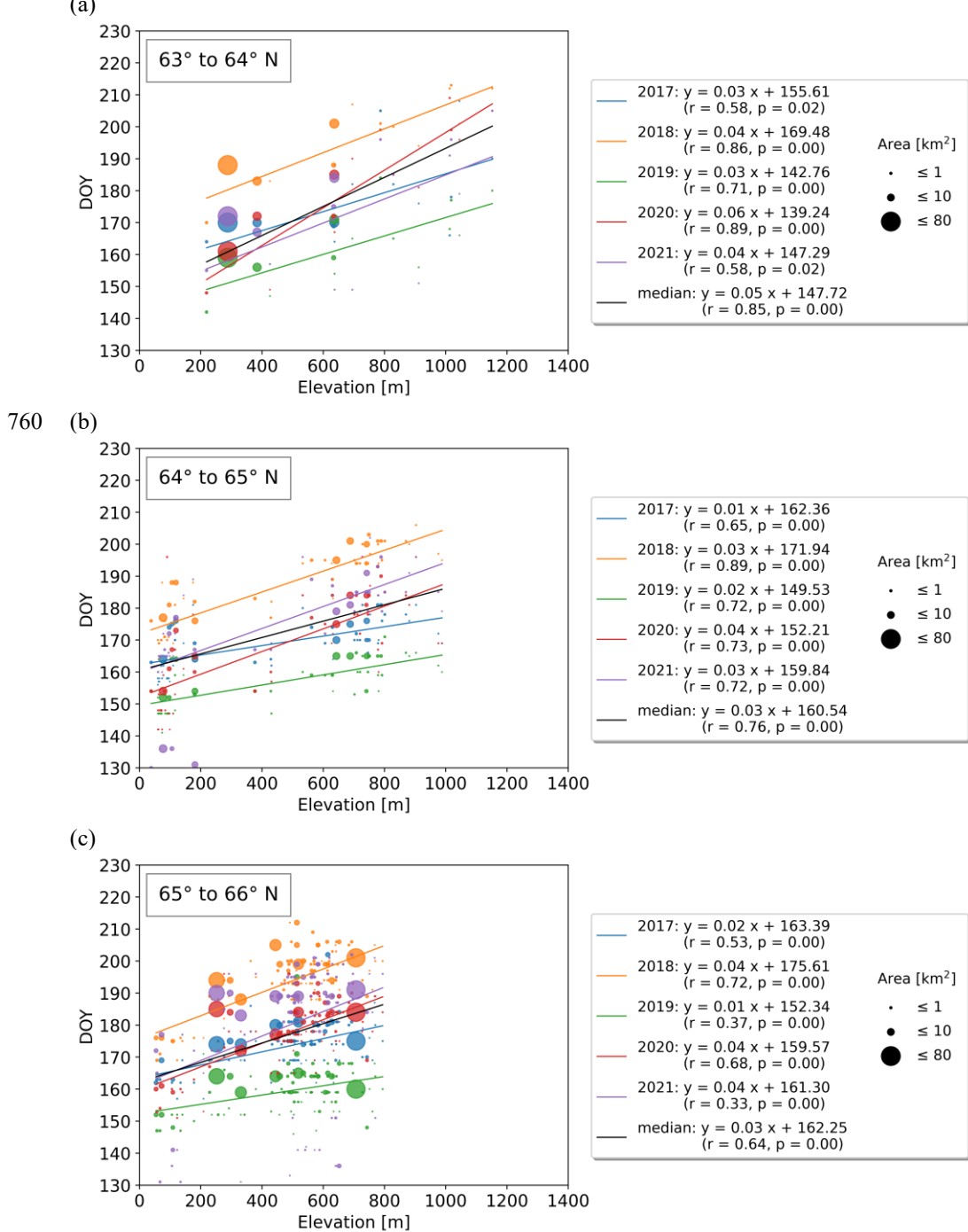

**Figure C3: (a)-(c) Yearly lake ice break-up timings vs elevation grouped by 1° latitude between 63° and 66° N. Several years exhibit strong correlations between break-up timing and elevation which increase by 3-6 days per 100 m elevation gain. Significance values of p = 0.00 in the figures indicate highly significant relations of p < 0.01.**

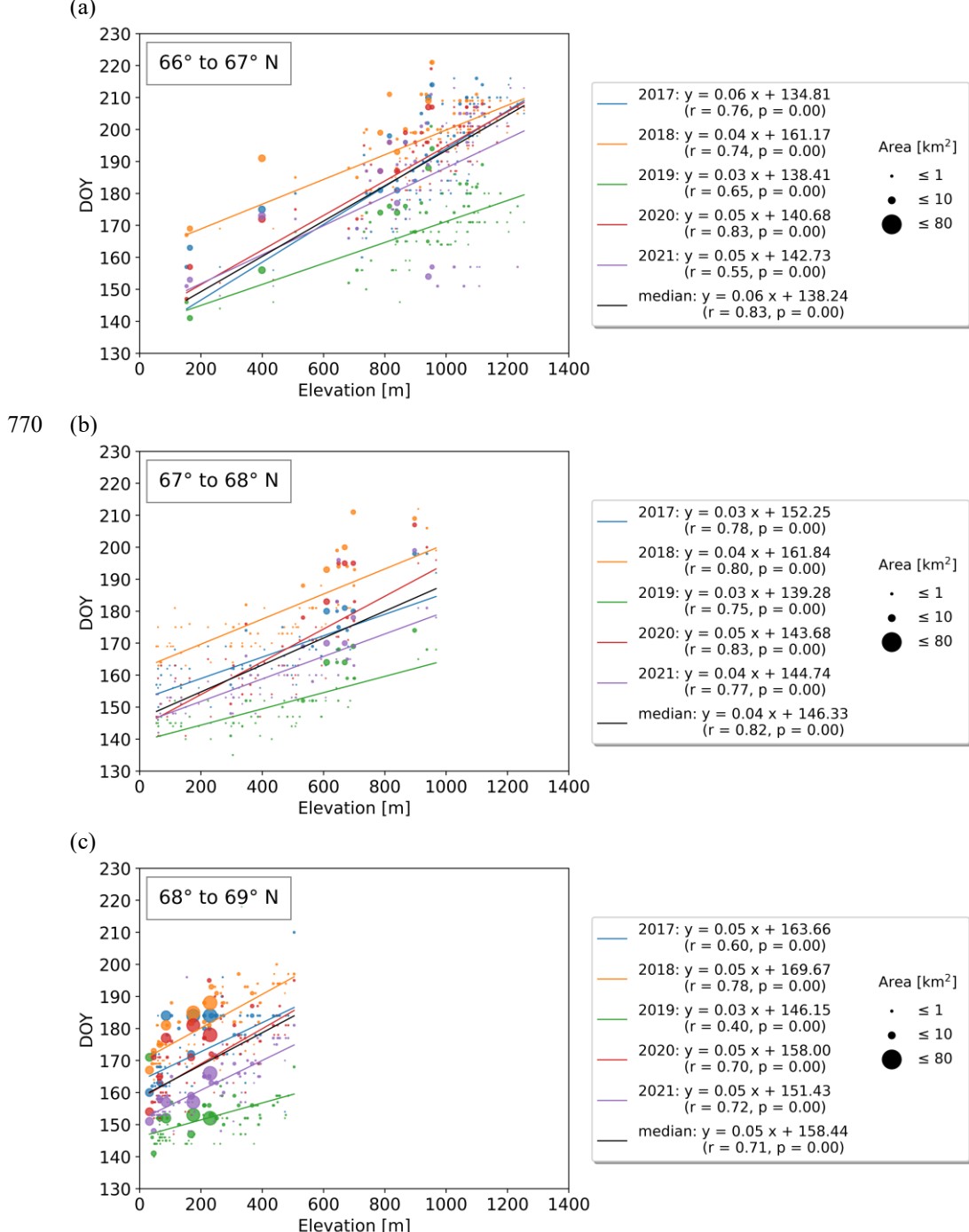

**Figure C4: (a)-(c) Yearly lake ice break-up timings vs elevation grouped by 1° latitude between 66° and 69° N. Several years exhibit strong correlations between break-up timing and elevation which increase by 3-6 days per 100 m elevation gain. Significance values of p = 0.00 in the figures indicate highly significant relations of p < 0.01.**

(a)

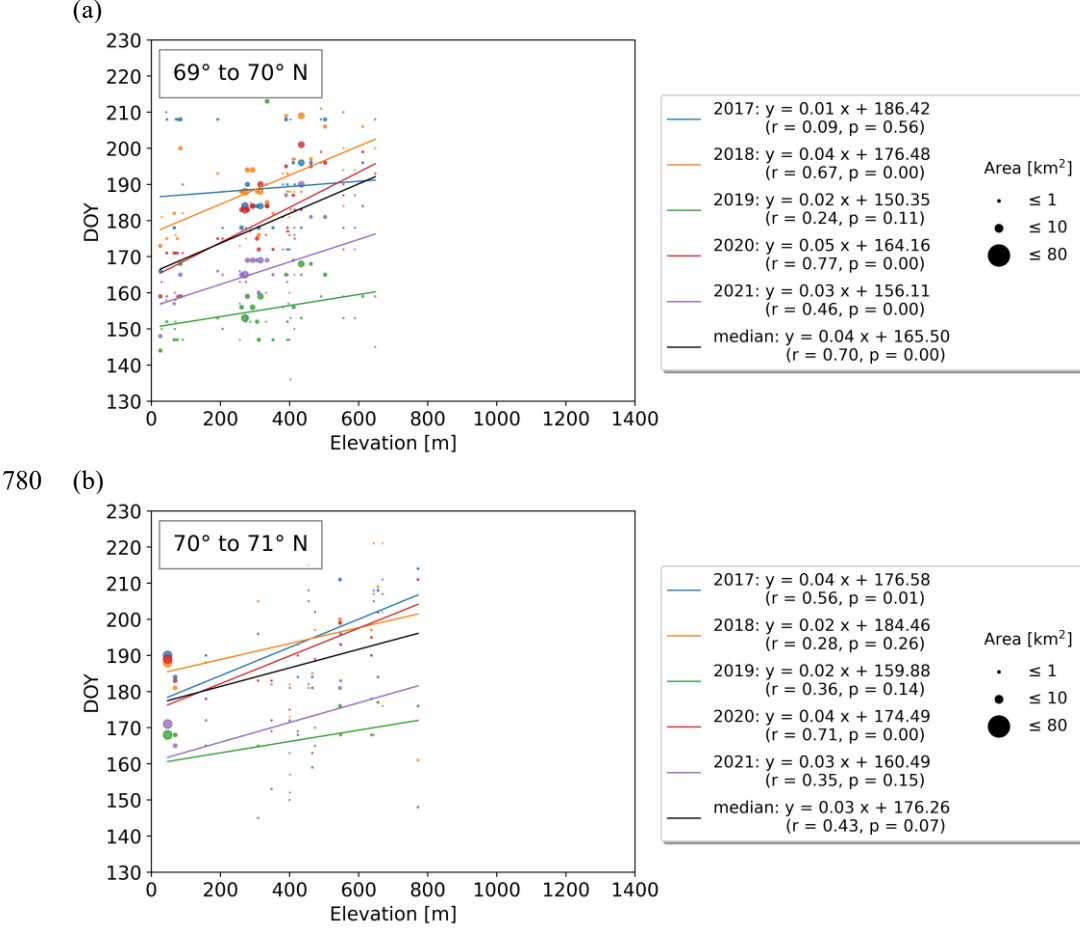

(b)

Figure C5: (a)-(c) Yearly lake ice break-up timings vs elevation grouped by 1° latitude between 69° and 71° N. Several years exhibit strong correlations between break-up timing and elevation which increase by 3-5 days per 100 m elevation gain. Significance values of p = 0.00 in the figures indicate highly significant relations of p < 0.01.


**Table C1: Correlations characteristics of the relation between lake ice break-up timing and latitude as well as between lake ice break-up timing and lake size. The number of relations N for assessing the number of significant relations n corresponds to the number of years (lake-specific annual break-up DOY vs. latitude/lake size), to the number of elevation bands (lake-specific 2017-2021 median break-up DOY vs. latitude/lake size), to the number of latitude bands (lake-specific 2017-2021 median break-up DOY vs. lake size), to years multiplied by the number of elevation bands (lake-specific annual break-up DOY and latitude/lake size grouped by elevation bands of 100 m) or to years multiplied by the number of latitude bands (lake-specific annual break-up DOY and lake size grouped by latitude bands of 1°). Correlations based on lake-specific 2017-2021 median break-up DOY as well as the low number of significant relations in most cases highlight the weak relationship between lake ice break-up timing and latitude or lake size.**

| Correlation between | Correlation coefficient r | Significance value p | Being significant (p ≤ 0.05) in n/N relations |
|---|---|---|---|
| Lake-specific 2017-2021 median break-up DOY and latitude | -0.18 | < 0.01 | 1/1 |
| Lake-specific 2017-2021 median break-up DOY and latitude (lakes grouped by elevation bands of 100 m) | $0.32 \leq r \leq 0.45$ | ≤ 0.05 | 7/13 |
| Lake-specific annual break-up DOY and latitude | $-0.41 \leq r \leq 0.16$ | < 0.01 | 5/5 |
| Lake-specific annual break-up DOY and latitude (lakes grouped by elevation bands of 100 m) | $-0.41 \leq r \leq 0.58$ | ≤ 0.05 | 21/65 |
|  |  |  |  |
| Lake-specific 2017-2021 median break-up DOY and lake size | 0.03 | 0.47 | - |
| Lake-specific 2017-2021 median break-up DOY and lake size (lakes grouped by elevation bands of 100 m) | $0.25 \leq r \leq 0.58$ | ≤ 0.05 | 6/13 |
| Lake-specific 2017-2021 median break-up DOY and lake size (lakes grouped by latitude bands of 1°) | 0.50 | 0.05 | 1/11 |
| Lake-specific annual break-up DOY and lake size | - | - | 0/5 |
| Lake-specific annual break-up DOY and lake size (lakes grouped by elevation bands of 100 m) | $-0.40 \leq r \leq 0.65$ | ≤ 0.05 | 25/65 |
| Lake-specific annual break-up DOY and lake size (lakes grouped by latitude bands of 1°) | $-0.20 \leq r \leq 0.60$ | ≤ 0.05 | 5/55 |

**Appendix D**

(a)                                                              (b)

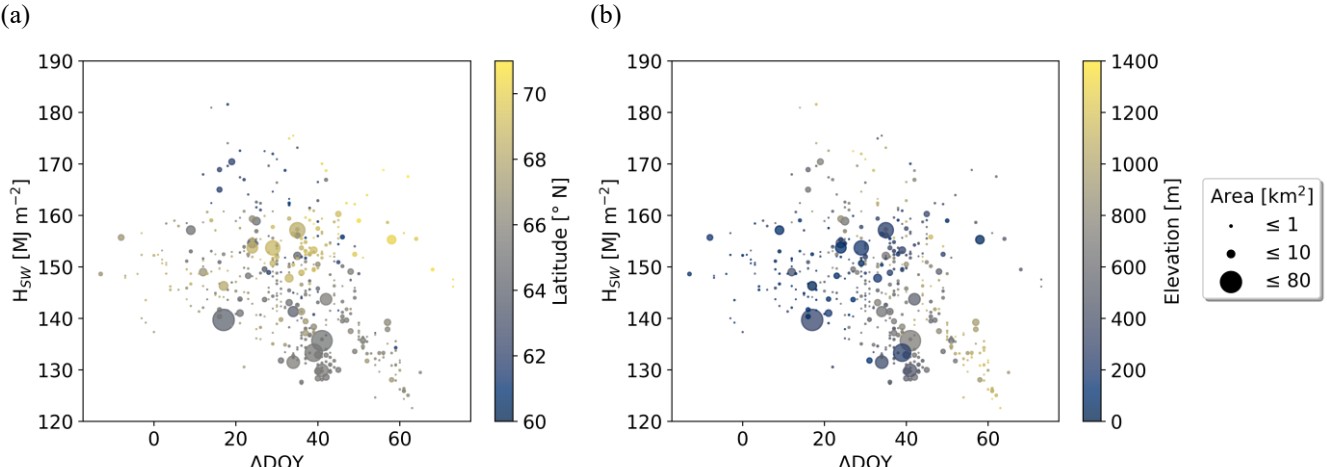

Figure D1: Excess radiation $H_{SW}$ due to lake-specific break-up timings which are 8 days earlier compared to the median break-up timings for the period 2017 to 2021 vs the time lag between the date of maximum incoming solar radiation and median lake ice break-up (ΔDOY) with color signatures for (a) latitude and (b) elevation. The median value of ΔDOY for the studied lakes amounts to 35 days (i.e., the lake-specific median break-up timing being 35 days after the solar radiation maximum).

(a)                                                              (b)

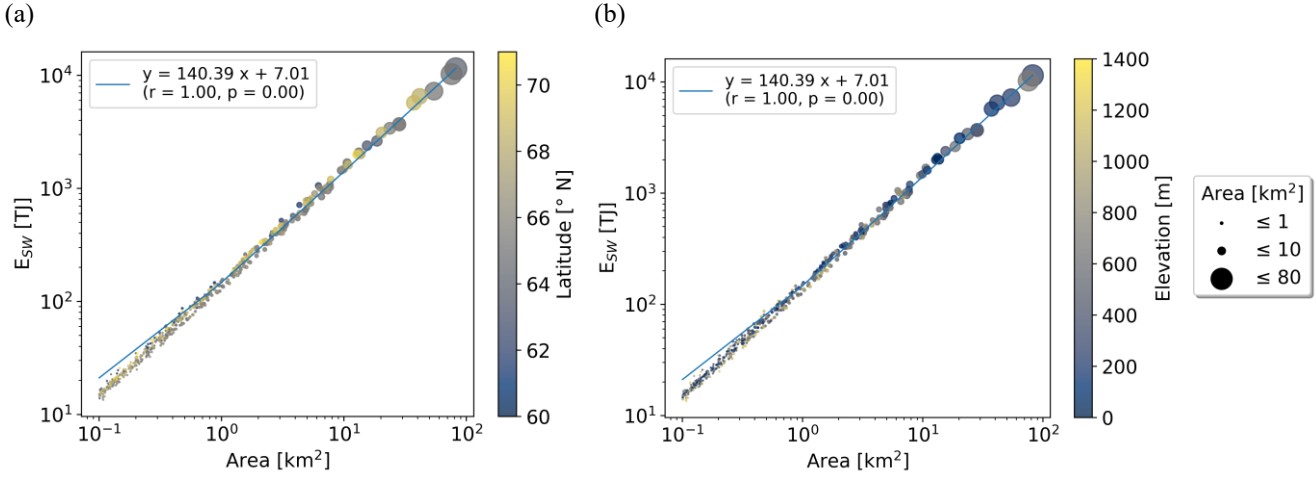


Figure D2: Excess energy $E_{SW}$ due to lake-specific break-up timings which are 8 days earlier compared to the median break-up timings for the period 2017 to 2021 vs lake surface area with color signatures for (a) latitude and (b) elevation. Lake surface areas strongly determine $E_{SW}$ values explaining more than 99 % of the variability in $E_{SW}$ (p < 0.01).