# Peer review of "Lake Ice Break-Up in Greenland: Timing and Spatio-Temporal Variability"

_EGUsphere, 2023_

## Author Response (AR1)

**RC1: 'Comment on egusphere-2023-1762', Anonymous Referee #1, 02 Sep 2023**

Dear editor, dear referees,
we are very grateful for two very constructive reviews and the editorial advice and appreciate the valuable time put into this. We believe by incorporating the reviews in our reviewed manuscript we achieved a much more mature manuscript.
In the following we mark **bold** the comments given by the referees and show the respective revisions in our manuscript in *italic*.
Once again, many thanks for the valuable input and all the best,
Christoph, on behalf of the author team

**This study uses Sentinel-1 (S1) combined with surface backscatter differences to obtain disintegration dates for 563 lake ice in Greenland between 2017-2021, obtaining a relationship between disintegration date and altitude. Estimates of the additional radiative input due to the advancement of the disintegration dates are presented, and the possible consequences of the excess energy are discussed. The results reveal the uniqueness of Greenland's geographic location and topographic features in response to climatic conditions, and enhance understanding of the extent to which altitude, latitude and lake size correlate with disintegration dates.**

**However, more detailed descriptions should be added in some areas, e.g. the introduction clarifies the importance of lake ice phenology studies but lacks a description of climate studies or ice condition studies in Greenland. In addition, why did the authors only analyse the date of disintegration and ignore the date of freezing, were they limited by the detection method? It would be more convincing to quantify the additional heat input in terms of spatial and temporal variations in the length of the complete freeze-up. For the lack of credibility of the authors' assumption of a pre-melt lake ice albedo $\alpha i$ = 0.9, which is perhaps only achievable with fresh snow; the effect of cloudiness on incident solar radiation should also be considered.**

**The research work is interesting and covers a large number of lakes in Greenland, and the authors are advised to improve the manuscript in two ways. ①, emphasise the climatic characteristics of Greenland and the progress of ice condition research, suddenly the significance of this study; ②, determine empirical values of regional lake ice albedo and cloudiness to ensure that the estimation of additional heat input is accurate. In addition (③), it would have been better to include freezing dates in the analyses, or to explain the reasons for not considering freezing date identification. Below are some specific comments aimed to provide guidance on the revision.**

*Thank you, we greatly appreciate the constructive remarks and we tried to incorporate the raised points in our revised manuscript. We address the remarks in detail below the specific comments.*

**L9-10: "There is a strong correlation between break-up date and elevation, while no relationship with latitude and lake area could be observed" 'no relationship' is too absolute, suggest 'weakly relationship'.**

*L9-10: Changed to "There is a strong correlation between break-up date and elevation, while a weak relationship with latitude and lake area could be observed."*

*L785-795: Added "Table C1: Correlations characteristics of the relation between lake ice break-up timing and latitude as well as between lake ice break-up timing and lake size. The number of relations N for assessing the number of significant relations n corresponds to the number of years (lake-specific annual break-up DOY vs. latitude/lake size), to the number of elevation bands (lake-specific 2017-2021 median break-up DOY vs. latitude/lake size), to the number of latitude bands (lake-specific 2017-2021 median break-up DOY vs. lake size), to years multiplied by the number of elevation bands (lake-specific*

[revised manuscript text omitted]

**L21: "The duration of lake ice controls......" 'Duration of lake ice' might make more sense than 'break-up time'.**

*The SAR backscatter data does not allow for identifying the timing of freeze-up, this is why we could not focus on lake ice duration.*

*L306-407: Added "5.1 Limitations and Potentials*

*In our study, we apply a dynamic numerical threshold on the annual SAR backscatter evolution to establish an automated detection of lake ice break-up timing. Differences in the dielectric properties between water and ice which are manifested in a decline in radar backscatter (Unterschultz et al., 2009) and the fast transition from an ice-covered to an open water surface allow for an automated detection. In contrast, different modes of lake surface freeze-up as well as high local and inter-annual variability of backscatter during long periods of gradual freezing makes an automated detection of freeze-up very challenging. This limitation in the automated detection method constrains this study to an analysis of the variability of lake ice break-up timing as opposed to an analysis of spatial and temporal variations in the length of the period in which the lake surface is frozen. However, we argue, that regarding the impact of lake ice presence on surface energy balance, the break-up timing of lake ice is more relevant than the timing of freeze-up. In our data we see that freeze-up typically occurs in October or November when incoming solar radiation is lower than during the timing of break-up (yearly median break-up timings vary between 8 June and 10 July) (Fig 2)."*

**L34-35: "The seasonal changes in solar radiation, however, are the main influence for the overall energy availability to form and decay lake ice cover" It is for this reason that solar radiation may have a greater effect on lake ice melt than air temperature. Does it make sense to study the use of "Calculating Cumulative Positive Degree Days"?**

*We added air temperature (cumulative PDD) to our analysis since we assumed its significance based on several other studies which showed air temperature strongly determining the break-up timing (or duration). Studies by Magnuson et al. (2000), Weyhenmeyer et al. (2004), Williams et al. (2004), Duguay et al. (2006), Korhonen (2006), Williams and Stefan (2006), Brown and Duguay (2010), Jeffries et al. (2012), and Imrit and Sharma (2021) show both linear and non-linear relations between lake ice break-up and air temperature, as well as February-March and April-May air temperatures being*

*determining lake ice break-up factors. However, we did not aim to quantify statistical relationships between air temperature and break-up timing but rather discuss the complexity regarding factors such as latitude, elevation, radiation and PDDs.*

*L32-38: Changed to "Lake ice freeze-up and break-up are results of energy surplus or deficit in the energy balance of the lake. The energy exchanges between the ice cover or water surface and the atmosphere are mainly determined by air temperature, precipitation, wind and radiation. The seasonal changes in solar radiation, however, are the main influence for the overall energy availability to form and decay lake ice cover (Brown and Duguay, 2010). Both linear and non-linear relation between lake ice break-up timing and air temperature have been established, while stronger correlations with latitude were identified compared to elevation (Magnuson, et al., 2000; Weyhenmeyer et al., 2004; Williams et al., 2004; Duguay et al, 2006; Korhonen, 2006; Williams and Stefan, 2006; Brown and Duguay, 2010; Jeffries et al., 2012; Imrit and Sharma, 2021)."*

*L216-220: Changed to "In order to understand the relationship between the annual evolution of air temperature, incoming radiation and median lake ice break-up timings, we calculate the climatological mean of positive degree days (PDDs) from RACMO2.3p2 2 m daily air temperature averages for the period 1991 to 2020 and analyze them as cumulative values from 1 January until the median DOY of break-up. With this we support our discussion on the complexity regarding determining factors such as latitude, elevation, and radiation in context with cumulative PDDs."*

**L43: "Wang et al., 2018" missing information in References**

*L694-695: Added "Wang, J., Duguay, C. R., Clausi, D. A., Pinard, V., and Howell, S. E. L.: Semi-Automated Classification of Lake Ice Cover Using Dual Polarization RADARSAT-2 Imagery, Remote Sens., 10(11), 1727, 1-27, https://doi.org/10.3390/rs1011172, 2018."*

**L63-65: "While still being dependent on field measurements for validation, lake ice studies from remote sensing are no longer dependent solely on a small number of ground observations but can produce results and extrapolate measurements across landscapes and regions" I didn't understand the sentence.**

*L87-89: Changed to "Lake ice studies from remote sensing can produce results and extrapolate field measurements across large spatial scales as opposed to field studies based on a small number of ground observations. However, field observations are pivotal for validation purposes. (Murfitt and Duguay, 2021)."*

**L91-92: "…below 2 days for most of Greenland" This is a great idea, has anyone else also achieved 2 days accuracy with the help of this approach?**

*Thanks for the kind words! In L75-97 we provide an overview of which studies are focusing on lake ice phenology based on remote sensing and address which data they used. To our knowledge, our study is the first study focusing on lake phenology in Greenland on a large scale and achieving that high temporal resolution.*

**L117-118: "…we acquired incoming shortwave radiation and air temperature at 2 m data as climatological daily mean values for the period 1991-2020 from RACMO2" The study used "daily mean values". The credibility of Cumulative Positive Degree Days can be enhanced if there are measured daily mean values between 2017-2022.**

*Since we attempt to cover a large spatial scale with high spatial variability in atmospheric variables, a seamless product from a high latitude adapted regional climate model (RCM) is an advantage.*

*RACMO2.3p2 is a high-resolution RCM that has been shown to capture spatial and temporal variability and absolute values well (Noël et al., 2019).*

*L139-152: Changed to "We use the current operational version of the Regional Atmospheric Climate Model (RACMO2.3p2), which is a high-resolution regional climate model (RCM) adapted for high latitudes. It shows to capture spatial and temporal variability and absolute values well (Noël et al., 2019). Air temperature and shortwave radiation data from RACMO2.3p2 is used to bring lake-specific climatological variables into context with lake ice break-up timing. We utilize model outputs as opposed to measurements since field observations do not possess the spatial coverage for our large-scale study. The current operational version RACMO2.3p2 is validated against 37 automated weather stations (AWSs) on the GIS and proves to realistically represent near-surface temperature (0.73 < R2 < 0.98) and cloud conditions through shortwave and longwave radiation components (0.85 < R2 < 0.96). This translates to biases in daily mean air temperatures at 2 m and incoming shortwave radiation of 0.14 °C and 4.8 W m-2, the latter corresponding to a bias of 2.7 % (Noël et al., 2019).*

*For estimating excess radiation and energy due to variability in the break-up timing and investigating potential correlations with air temperature, we acquire lake-specific incoming shortwave radiation at the surface and air temperature at 2 m data as climatological daily mean values for the period 1991-2020 from RACMO2.3p2 (Noël et al., 2019. The four nearest grid points of the model from the coordinates of each respective lake center point are selected in order to approximate the radiation and temperature data using Delaunay triangulation (Delaunay, 1934) with cubic interpolation."*

**L132-133: "…when most of the lake surface is…" "Most" is not a quantitative description; 80% or 90% would be a better standard.**

*We rephrased the paragraph highlighting the nature of our method. However, we cannot state a definite quantitative measure since we do not perform a pixel-based ice cover classification (which would be beyond the scope and computing power at a that large sample we are working with).*

*L164-172: Changed to "The term "lake ice break-up" used in this study describes the timing, i.e., day of year (DOY), when at least 80 % of the lake surface is liquid water and is therefore an approximation to the timing of "water clear of ice" (WCI) (WMO et al., 2023b). Once snowmelt starts on lake ice, water collects around the margins where it warms as it absorbs solar radiation and accelerates melting due to positive feedback (Jeffries et al., 2012). We assume that lake ice is longest present in the central areas of the lake and therefore aim to detect the presence or absence of ice in the central 20 % of the lake surface area which means that the $\sigma\_0$ values are averaged for this central portion. This results in an area of approximately 0.02 km2 for the smallest lake, which corresponds to at least 200 pixels considered for averaging and proves to be a robust measure to identify the phenological state of lake ice."*

**L150: "Figure 1 (a)" could the sudden rise in November $\sigma$ also determine the freezing of the lake?**

*Yes, we also assume that this period represents the freezing of the exemplary lake.*

*L306-407: Added "5.1 Limitations and Potentials*

*In our study, we apply a dynamic numerical threshold on the annual SAR backscatter evolution to establish an automated detection of lake ice break-up timing. Differences in the dielectric properties between water and ice which are manifested in a decline in radar backscatter (Unterschultz et al., 2009) and the fast transition from an ice-covered to an open water surface allow for an automated detection. In contrast, different modes of lake surface freeze-up as well as high local and inter-annual variability of backscatter during long periods of gradual freezing makes an automated detection of freeze-up very challenging. This limitation in the automated detection method constrains this study to an analysis of*

*the variability of lake ice break-up timing as opposed to an analysis of spatial and temporal variations in the length of the period in which the lake surface is frozen. However, we argue, that regarding the impact of lake ice presence on surface energy balance, the break-up timing of lake ice is more relevant than the timing of freeze-up. In our data we see that freeze-up typically occurs in October or November when incoming solar radiation is lower than during the timing of break-up (yearly median break-up timings vary between 8 June and 10 July) (Fig 2)."*

**L150: "Figure 1 I" What does the pentagram represent?**

*L191-197: Added pentagram to the legend in Figure 1*

**L161-162: "The progressing melt on the lake surface leading to a rougher, wetter surface explains the $\sigma0$ recovery before the major backscatter decline in summer indicating lake ice break-up" rougher and wetter ice is unlikely to result in α = 0.9.**

*While it is true that the bare lake ice albedo might be as low as 0.2-0.5 in its latest stage before break-up (Mullen and Warren 1988, Heron and Woo 1994, Henneman and Stefan 1999, Grenfell and Perovich 2004, Jakilla et al. 2009, Semmler et al. 2012, Svacina et al. 2014a, Svacina et al. 2014b, Leppäranta 2015, Zdorovennova et al. 2018, and Robinson et al. 2021), we acknowledge that the hypothesized 8-day-earlier break-up must not only take the latest stage of the ice cover into account but surface albedos along the melt season (dry snow, wet snow, bare). We hypothesize that an 8-days-earlier break-up translates to the period of snow-covered lake surface being 8 days shorter and the period of open water being 8 days longer, while the length of the period of disintegration is being maintained. Therefore, we base our excess energy estimates on a change from snow albedo to open water albedo.*

*L249-260: Changed to "We hypothesize that an earlier lake ice break-up impacts the timing of the entire lake ice disintegration process from the melting of the initially dry snow cover on top of the lake ice until the melting of the bare ice surface itself. The short-term albedo development might be highly variable, impacting the transition from dry to wet snow, from wet snow to bare ice, and from bare ice to open water. We assume that the earlier break-up exhibits a shift of the entire disintegration period while its length is maintained, which means that the period of the snow-covered lake is 8 days shorter while the period of the lake having an open water surface is 8 days longer. Therefore, the albedo difference Δα is expressed as the change from snow-covered lake ice α_i (0.9) to open water α_w (0.1) to quantify the excess radiation input H_SW."*

*L416-422: Added "The estimates of the excess radiation and consequently excess energy are based on assuming a shift of the entire break-up process while having its length maintained. This means that the excess energy is assumed to be available for the open water surface of the lake and is not utilized during the break-up process. Short-time albedo development during the disintegration process of the lake surface might be highly variable and changes in the configurations and lengths of the transition from dry to wet snow, from wet snow to bare ice and from bare ice to open water might vary greatly with elevation, latitude and local climates. A site-specific characterization of albedo evolution during lake ice break-up might be a subject for further studies and improve local excess energy estimates."*

**L164-165: "This is due to the nature of break-up processes being more complex due to melting on top and bottom or varying acquisition conditions" could it be a secondary freeze due to lower air temperatures?**

*Assessing optical satellite imagery, we are very confident that in this period no secondary freeze happened. We also did not see any secondary freeze events after the major break-ups in our data.*

**L179: "Calculating Cumulative Positive Degree Days" whether climate averages for the period 1991-2020 are representative of today's ice-season environmental conditions?**

*We acknowledge that the PDD from 2017-2021 is going to be higher compared to the 1991-2020 period due to the fact that the region is warming quickly with respect to the 1991-2020 period (Hanna et al., 2021). We utilized the 1991-2020 climatologies to get a robust comparison to the median break-up timings between 2017 and 2021. We expected that a yearly comparison to cum. PDDs or to average cum. PDDs for the period 2017-2021 are not representative of a general lake-specific generalization due to the exceptional years of 2018 and 2019 (Hanna et al. 2021).*

**L214: "…of lake ice α (0.9) …" the value of 0.9 is too high, please revise or give a basis.**

*While it is true that the bare lake ice albedo might be as low as 0.2-0.5 in its latest stage before break-up (Mullen and Warren 1988, Heron and Woo 1994, Henneman and Stefan 1999, Grenfell and Perovich 2004, Jakilla et al. 2009, Semmler et al. 2012, Svacina et al. 2014a, Svacina et al. 2014b, Leppäranta 2015, Zdorovennova et al. 2018, and Robinson et al. 2021), we acknowledge that the hypothesized 8-day-earlier break-up must not only take the latest stage of the ice cover into account but surface albedos along the melt season (dry snow, wet snow, bare). We hypothesize that an 8-days-earlier break-up translates to the period of snow-covered lake surface being 8 days shorter and the period of open water being 8 days longer, while the length of the period of disintegration is being maintained. Therefore, we base our excess energy estimates on a change from snow albedo to open water albedo.*

*L249-260: Changed to "We hypothesize that an earlier lake ice break-up impacts the timing of the entire lake ice disintegration process from the melting of the initially dry snow cover on top of the lake ice until the melting of the bare ice surface itself. The short-term albedo development might be highly variable, impacting the transition from dry to wet snow, from wet snow to bare ice, and from bare ice to open water. We assume that the earlier break-up exhibits a shift of the entire disintegration period while its length is maintained, which means that the period of the snow-covered lake is 8 days shorter while the period of the lake having an open water surface is 8 days longer. Therefore, the albedo difference Δα is expressed as the change from snow-covered lake ice α_i (0.9) to open water α_w (0.1) to quantify the excess radiation input H_SW."*

*L416-422: Added "The estimates of the excess radiation and consequently excess energy are based on assuming a shift of the entire break-up process while having its length maintained. This means that the excess energy is assumed to be available for the open water surface of the lake and is not utilized during the break-up process. Short-time albedo development during the disintegration process of the lake surface might be highly variable and changes in the configurations and lengths of the transition from dry to wet snow, from wet snow to bare ice and from bare ice to open water might vary greatly with elevation, latitude and local climates. A site-specific characterization of albedo evolution during lake ice break-up might be a subject for further studies and improve local excess energy estimates."*

**L223-224: Does the authors take into account the effect of cloud cover, which attenuates incident shortwave radiation?**

*L143-147: Changed to "The current operational version RACMO2.3p2 is validated against 37 automated weather stations (AWSs) on the GIS and proves to realistically represent near-surface temperature ($0.73 < R^2 < 0.98$) and cloud conditions through shortwave and longwave radiation components ($0.85 < R^2 < 0.96$). This translates to biases in daily mean air temperatures at 2 m and incoming shortwave radiation of 0.14 °C and 4.8 W m$^{-2}$, the latter corresponding to a bias of 2.7 % (Noël et al., 2019)."*

**L280: "Figure 4 (a)" I want to know r = 0.76, then p-value = ?**

*We included probability values for all reported statistical relationships.*

**L284: "…we there is no relationship between break-up timing and lake size" it is necessary to control for equal elevation and latitude, and to analyse only the size of the lake in order to consider that there is no relationship between disintegration and lake size**

*We analyzed the lake size by elevation and latitude bands, however, in order to keep the manuscript and figures concise and readable, we did not include all details in the figures. We will add the requested information and expand on this in the main text.*

*L208-210: Changed to "Furthermore, we group lakes into sections of 1° N latitude and 100 m elevation, respectively, to assess spatial gradients and explore relationships between break-up timing and elevation, latitude as well as lake surface area."*

*L337-341: Changed to "Subdivided into latitudinal bands of 1° between 60° N and 71° N, strong correlations (up to r = 0.89, p < 0.01) between break-up timing and elevation can be identified in several years. Those exhibit an increase of 3-6 DOY per 100 m depending on the latitudinal band as well as the elevation range and are significant in 43 out of the 55 yearly correlations (Fig. C2-Fig. C5). The median break-up dates for the period 2017 to 2021, except between 60-61° N and 70-71° N, show strong correlations (0.64 ≤ r ≤ 0.85, p ≤ 0.01) increasing by 3-6 DOY per 100 m elevation increase."*

*L785-795: Added "Table C1: Correlations characteristics of the relation between lake ice break-up timing and latitude as well as between lake ice break-up timing and lake size. The number of relations N for assessing the number of significant relations n corresponds to the number of years (lake-specific annual break-up DOY vs. latitude/lake size), to the number of elevation bands (lake-specific 2017-2021 median break-up DOY vs. latitude/lake size), to the number of latitude bands (lake-specific 2017-2021 median break-up DOY vs. lake size), to years multiplied by the number of elevation bands (lake-specific annual break-up DOY and latitude/lake size grouped by elevation bands of 100 m) or to years multiplied by the number of latitude bands (lake-specific annual break-up DOY and lake size grouped by latitude bands of 1°). Correlations based on lake-specific 2017-2021 median break-up DOY as well as the low number of significant relations in most cases highlight the weak relationship between lake ice break-up timing and latitude or lake size.*

| Correlation between | Correlation coefficient r | Significance value p | Being significant (p ≤ 0.05) in n/N relations |
|---|---|---|---|
| *Lake-specific 2017-2021 median break-up DOY and latitude* | *-0.18* | *< 0.01* | *1/1* |
| *Lake-specific 2017-2021 median break-up DOY and latitude (lakes grouped by elevation bands of 100 m)* | *0.32 ≤ r ≤ 0.45* | *≤ 0.05* | *7/13* |
| *Lake-specific annual break-up DOY and latitude* | *-0.41 ≤ r ≤ 0.16* | *< 0.01* | *5/5* |
| *Lake-specific annual break-up DOY and latitude (lakes grouped by elevation bands of 100 m)* | *-0.41 ≤ r ≤ 0.58* | *≤ 0.05* | *21/65* |
| | | | |
| *Lake-specific 2017-2021 median break-up DOY and lake size* | *0.03* | *0.47* | *-* |
| *Lake-specific 2017-2021 median break-up DOY and lake size (lakes grouped by elevation bands of 100 m)* | *0.25 ≤ r ≤ 0.58* | *≤ 0.05* | *6/13* |
| *Lake-specific 2017-2021 median break-up DOY and lake size (lakes grouped by latitude bands of 1°)* | *0.50* | *0.05* | *1/11* |

| Lake-specific annual break-up DOY and lake size | - | - | 0/5 |
|---|---|---|---|
| Lake-specific annual break-up DOY and lake size (lakes grouped by elevation bands of 100 m) | -0.40 ≤ r ≤ 0.65 | ≤ 0.05 | 25/65 |
| Lake-specific annual break-up DOY and lake size (lakes grouped by latitude bands of 1°) | -0.20 ≤ r ≤ 0.60 | ≤ 0.05 | 5/55 |

"

**L290-291: "Figure 5b shows that lakes with similar cumulative PDDs experience a later lake ice break-up at higher elevation" are the air temperature data corresponding to lakes at different elevations at the same latitude the same? Because altitude must bring about differences in air temperature, defaulting to the same value will lead to bias.**

*L148-152: Changed to "For estimating excess radiation and energy due to variability in the break-up timing and investigating potential correlations with air temperature, we acquire lake-specific incoming shortwave radiation at the surface and air temperature at 2 m data as climatological daily mean values for the period 1991-2020 from RACMO2.3p2 (Noël et al., 2019. The four nearest grid points of the model from the coordinates of each respective lake center point are selected in order to approximate the radiation and temperature data using Delaunay triangulation (Delaunay, 1934) with cubic interpolation."*

**L332: "…down to 35 m by 1 K across…" 35 m is the average depth of regional lakes?**

*L281-287: Added "Our calculations show that lake surface area $A\_i$ strongly determines the excess energy input $E\_SW$ and explains more than 99 % of its variability in the dataset (Fig. D2). This allows for translating the excess energy input $E\_SW$ to ice thickness melted or water depth warmed by 1 K across the respective lake surface area which are derived the from excess radiation input $H\_SW$. For this estimate we ignore lake bathymetry and present the mean values of melted thickness of ice at the melting point $h\_i$ [m] and depth of water $h\_w$ [m] warmed by 1 K, as shown in Eq. (8) and Eq (9).*

$$h_i = mean(\frac{H_{SW}}{L_f \, \rho_i}) \qquad\qquad (8)$$
$$h_w = mean(\frac{H_{SW}}{c_w \, \rho_w}) \qquad\qquad (9)"$$

*L386-389: Changed to "Referring the added energy in terms of ice melt and water temperature rise to the lake-specific areas in a simplified way which ignores lake bathymetry, we find that the excess energy input averagely corresponds to melting 0.4 ± 0.1 m thick ice or heating up a water depth down to 35 ± 3 m by 1 K across the entire surface areas."*

**L349: "(between 40°N and 82.5°N)" perhaps because Greenland's latitudinal span is not large enough to become only weakly correlated.**

*L453-467: Added "5.3 Lake Ice Break-Up in Greenland compared to other Study Sites*

*L'Abée-Lund et al. (2021) studied the phenology of 101 Norwegian lakes between 1890-2020 covering a latitudinal range of 58.2-69.9° N and found stronger correlations between the average timing of ice break-up and elevation (3.4 days per 100 m, r = 0.63, p < 0.01) when compared to the correlations between break-up timing with latitude (2.3 days per 1° N, r = 0.35, p < 0.01.). Williams et al. (2004) and Williams and Stefan (2006) assessed lake ice break-up timing of approximately 140 lakes in North America (between 40.0° N and 82.5° N) from records between 1848 and 1997 and found that there is a strong relationship between break-up timing and latitude, arguing that geographic latitude is a good indicator of climate. They showed that only a weak relationship between elevation and the timing of*

*break-up was observed when grouping the data by region, presenting an increase of break-up timing of 2 days per 100 m elevation increase. Zhang et al. (2021) found a strong correlation between latitude and lake ice-break up timing (R2 = 0.75) and only a weak relationship with elevation for 4241 lakes in Alaska over the period 2000-2019.*

*In our study which covers lakes between 60° N and 71° N, however, we find strong correlations between break-up timing and elevation and only a weak relationship with latitude for lakes at similar elevation. This again highlights the influence of the spatial configuration of lakes in Greenland determined by the proximity to both the ocean and the GIS, the presence of fjord systems and the steep slopes."*

**RC2: 'Comment on egusphere-2023-1762', Anonymous Referee #2, 25 Sep 2023**

Dear editor, dear referees,
we are very grateful for two very constructive reviews and the editorial advice and appreciate the valuable time put into this. We believe by incorporating the reviews in our reviewed manuscript we achieved a much more mature manuscript.
In the following we mark **bold** the comments given by the referees and show the respective revisions in our manuscript in *italic*.
Once again, many thanks for the valuable input and all the best,
Christoph, on behalf of the author team

**General comment:**

**I enjoyed reading the paper. The text is clear. The quality of the plots are very good. I think it is a nice paper and contributes to the field.**

*Thank you, we greatly appreciate the constructive remarks and we tried to incorporate the raised points in our revised manuscript. We address the remarks in detail below the specific comments.*

**I have one general (major) comment:**

**I think the discussion is somewhat lacking in putting the work in the broader context. It is not well motivated why this study matters from a global perspective and what it adds to the general understanding of the climate change impacts. More specifically:**

- **US is a wide continent covering various Köppen-Geiger climatic zone but Greenland is specifically located in one climatic zone and is not spanning across multiple latitudes with different climatic features. Therefore, I am not too surprised that for US there was a strong spatial correlation while such correlation could not be observed in Greenland. Please consider adding more to this discussion and make the comparison a bit stronger.**
  *L432-442: Added "When discussing break-up timing in context of topography, the spatial distribution of snowfall must also be considered. The amount of snowfall may greatly influence the variability of lake ice due to its insulating properties on top of the lake ice surface as well its role during lake ice build-up and disintegration. The difference in break-up timing between coastal lakes which is earlier at 67.5-68.5° N compared to being later at > 68.5° N may be attributed to higher snow accumulation rates which can be as much as three times higher in the latter case, as shown by Bales et al., 2009.*
  *Imrit and Sharma (2021) found that climate change, warmer local air temperatures, and teleconnection patterns are able to explain on average approx. 60 % of the variation in ice phenology of lakes in Northern America and Europe. On average, approx. 40% of the variation remained unexplained and could be attributed to local weather conditions, such as solar radiation inputs, wind, snow cover and lake and landscape characteristics, such as lake depth, elevation and fetch. This is in line with our interpretation of variability in the timing of lake ice break-up which may be greatly attributed to local weather, climate and landscape characteristics in coastal Greenland."*

  *L452-466: Changed to "5.3 Lake Ice Break-Up in Greenland compared to other Study Sites*
  *L'Abée-Lund et al. (2021) studied the phenology of 101 Norwegian lakes between 1890-2020 covering a latitudinal range of 58.2-69.9° N and found stronger correlations between the average timing of ice break-up and elevation (3.4 days per 100 m, r = 0.63, p < 0.01) when compared to the correlations between break-up timing with latitude (2.3 days per 1° N, r = 0.35, p < 0.01.). Williams et al. (2004) and Williams and Stefan (2006) assessed lake ice break-up*

*timing of approximately 140 lakes in North America (between 40.0° N and 82.5° N) from records between 1848 and 1997 and found that there is a strong relationship between break-up timing and latitude, arguing that geographic latitude is a good indicator of climate. They showed that only a weak relationship between elevation and the timing of break-up was observed when grouping the data by region, presenting an increase of break-up timing of 2 days per 100 m elevation increase. Zhang et al. (2021) found a strong correlation between latitude and lake ice-break up timing (R2 = 0.75) and only a weak relationship with elevation for 4241 lakes in Alaska over the period 2000-2019.*

*In our study which covers lakes between 60° N and 71° N, however, we find strong correlations between break-up timing and elevation and only a weak relationship with latitude for lakes at similar elevation. This again highlights the influence of the spatial configuration of lakes in Greenland determined by the proximity to both the ocean and the GIS, the presence of fjord systems and the steep slopes."*

- **Put the whole study in the broader context: Maybe mention implication for hydropower. You can use snow-dominated locations for the sake of comparison. I tried to find some examples and the discussion section in paper https://www.sciencedirect.com/science/article/pii/S0022169423007497 and https://hess.copernicus.org/articles/24/3815/2020/ might work. Please try to find other papers to add to this point.**

*L494-514: Added "5.5 Implications of Lake Ice Break-Up Variability*

*The magnitudes of our estimated excess energies from earlier break-up timings indicate potentially vast changes in the energy balance with increasingly earlier break-up dates and increasing variabilities due to a changing climate. Furthermore, changes in lake ice phenology which influence sub-surface mixing, temperature and light conditions have a direct impact on the ecosystem. These ramifications due to an earlier lake ice break-up timing may manifest themselves in increases in hypolimnetic oxygen concentrations because of changes in the lake mixing regime, potential earlier and more $CO_2$ emissions by lakes during lake ice melting, earlier thermal stratification, a longer open-water season for warming, higher water temperatures, increased evaporation rates and potentially decreasing water levels. Furthermore, shifts in phytoplankton biomass, food web dynamics and community composition may occur due to unpredictable changes in lake mixing regimes (Imrit and Sharma, 2021). Below-ice aquatic biodiversity is often tied to the presence of ice, which in turn impacts the seasonal cycle of prey and predators, nutrient cycling, dissolved oxygen and the timing of algal blooms which may impact ecological processes even in summertime (Huang et al., 2022). Lake ice break-up timing may also have further downstream implications, such as freshwater input into fjords in regard to transport of nutrients and sediments which influence the geochemical composition and ultimately the marine primary production (Abermann et al., 2021).*

*Besides these direct impacts in natural systems, the variability in lake ice break-up exhibits several anthropogenic implications. Climate change is likely influencing the thickness of lake ice as well as the timing of breakup, which has a significant impact on both reservoir inflows and outflows for hydropower (Cherry et al., 2017) as well as on its infrastructure such as damns, spillways, channels, reservoirs, tunnels, inlets, and outlets (Gebre et al., 2013). Changes in phenological ice regimes will make access to lakes more uncertain and potentially hazardous and may impact traditional subsistence-based lifestyles which are dependent on the natural network for access to isolated communities, remote industrial developments, hunting, fishing, herding and trapping areas (Prowse et al., 2011)."*

- **Compare the finding with other boreal countries: Note https://tc.copernicus.org/articles/16/2493/2022/tc-16-2493-2022-discussion.html where they studied ice break-up patterns in Sweden. Please try to find other papers to add to this point.**

*L452-466: Changed to "5.3 Lake Ice Break-Up in Greenland compared to other Study Sites*

[revised manuscript text omitted]

*L525-528: Changed to "The variability in lake ice break-up timing of the studied lakes in Greenland for 2017-2021 is above the observed variability of related lake ice phenology studies in other regions over the last century which may be attributed to exceptional temperature years in recent years and the comparable short study period. However, with progressing climate change an increased interannual variability and earlier timing in lake ice break-up can be expected."*

**Specific comments:**

- **L280: please remove we.**
  *L335-338: Changed to "Figure 4: Median break-up timings $DOY\_m$ for the period 2017 to 2021 vs (a) elevation and (b) latitude. $DOY\_m$ increase by 3 DOY per 100 m elevation gain exhibiting a strong correlation (r = 0.76, p < 0.01) while only a weak correlation with latitude can be identified."*

- **Please report the significance of the correlations whenever you report the correlation strength.**
  *We included probability values for all reported statistical relationships.*

- **The paper lacks climatic description of Greenland.**
  *L51-74: Added "1.1 Climate in Coastal Greenland*
  *Greenland extends for approx. 23° of latitude, with temperature, precipitation and consequently mass balance rates varying considerably across latitudes and coasts (Westergaard-Nielsen et al., 2020; Hanna et al., 2021; Mankoff et al., 2021; Slater et al., 2021, Box et al., 2023). Due to the semi-permanent Icelandic Low and the rocky landscape, the Southeast coast receives particularly high amounts of precipitation (e.g., Ettema et al., 2010; Fettweis et al., 2017). As precipitation rates greatly decrease northward, North Greenland is classified as a polar desert with very shallow snow cover that quickly disappears in the warm season. Temperature also tends to decrease with latitude, related to snow and radiation conditions. However, other factors shape the coastal climate such as prominent ocean currents (e.g., East Greenland and North Atlantic current) as well as sea ice conditions (Westergaard-Nielsen et al., 2020). The West and East coasts also exhibit different topographic features, from a topographically complex Southeast contrasting with generally rather gentle slopes in Southwest or North Greenland (Karami et al., 2017). Nevertheless, both the East and West coasts comprise diverse fjord systems impacting regional climate and local wind conditions. Consequently, the leeward side of these inland mountain systems receive reduced precipitation.*

*Such coast-inland gradients are therefore complex, influencing the distribution of permafrost and freshwater systems (Westergaard-Nielsen et al., 2018; Abermann et al., 2021). While several studies on accumulation rates and rainfall exemplify the general East-West gradient in precipitation focusing on the Greenland Ice sheet (GIS) (e.g., Shen et al., 2012; Koenig et al., 2016, Box et al., 2023), Bales et al. (2009) include and highlight coastal variability of snow accumulation. In-situ and remote sensing data as well as polar-adapted climate models point to a Greenland-wide warming in recent decades, particularly in summer (e.g., Westergaard-Nielsen et al., 2018; Jiang et al., 2020; Hanna et al., 2021). Part of this warming is attributed to more frequent and intense anti-cyclonic conditions in the vicinity of Greenland, leading to advection of relatively warm air masses from low latitudes. Silva et al. (2022) showed that the warming applies to different circulation conditions. As a consequence of atmospheric warming, the ratio of liquid to total precipitation has increased in coastal areas particularly during summer (e.g., Huai et al. 2022; van der Schot et al. 2023). The Arctic Amplification is more pronounced during the cold season, with coastal temperature warming along the West coast linked with reduced sea ice in the Baffin Bay (e.g., Ballinger et al., 2021)."*

---

## Author Response (AR2)

Dear Dr. Yackel, reviewers and editorial team,

thank you very much for your constructive remarks and for your time and effort which helped us improving our manuscript and study.

We are happy and appreciate that our manuscript is now deemed ready for publication.

We also want to respond to the final suggestion of reviewer #2:

RACMO is forced by reanalyses, however, in our opinion this is not condition enough to categorize it as a reanalysis product. We follow the terminology of the providers (Noël et al., 2019, https://doi.org/10.1126/sciadv.aaw0123) and call it a polar-adapted regional climate model in our manuscript. For this reason, we did not revise this section.

The final version is now uploaded, in which we took the liberty to correct some last technical errors and add two lines regarding organizational requirements:

1) We corrected the wrong text font in line 172-180 from "caption" to "normal".

2) Due to our automated generation of the plots, statistical values in the low decimals were not represented properly which led to figures including "r = 1.00" and "p = 0.00". We corrected Fig. 4, C1, C2, C3, C4, C5, D2 to "r > 0.99" and "p < 0.01" to realistically represent these values.

3) We removed the empty line in the References in line 652.

4) We updated the Author contributions in line 541-542 and added "All authors contributed suggestions to the manuscript."

5) We added "The authors acknowledge the financial support by the University of Graz." to the Acknowledgements at line 546.

Thank you very much!

On behalf of the author team,

Christoph Posch